# METRIC-DRIVEN ATTRIBUTIONS FOR VISION TRANSFORMERS

**Chase Walker**[1], **Sumit Kumar Jha**[2], **Rickard Ewetz**[1]
[1] University of Florida
[2] Florida International University

## ABSTRACT

Attribution algorithms explain computer vision models by attributing the model response to pixels within the input. Existing attribution methods generate explanations by combining transformations of internal model representations such as class activation maps, gradients, attention, or relevance scores. The effectiveness of an attribution map is measured using attribution quality metrics. This leads us to pose the following question: if attribution methods are assessed using attribution quality metrics, why are the metrics not used to generate the attributions? In response to this question, we propose a Metric-Driven Attribution for explaining Vision Transformers (ViT) called MDA. Guided by attribution quality metrics, the method creates attribution maps by performing patch order and patch magnitude optimization across all patch tokens. The first step orders the patches in terms of importance and the second step assigns the magnitude to each patch while preserving the patch order. Moreover, MDA can provide a smooth trade-off between sparse and dense attributions by modifying the optimization objective. Experimental evaluation demonstrates the proposed MDA method outperforms 7 existing ViT attribution methods by an average of 12% across 12 attribution metrics on the ImageNet dataset for the ViT-base $16 \times 16$, ViT-tiny $16 \times 16$, and ViT-base $32 \times 32$ models. Code is publicly available at `https://github.com/chasewalker26/MDA-Metric-Driven-Attributions-for-ViT`.

## 1 INTRODUCTION

Computer vision models have been broadly adopted with applications in domains such as self-driving cars (Ando et al., 2023), surveillance (Ho et al., 2019), and disease detection (Esteva et al., 2021). However, due to the black-box nature of these models, proper explanations must be available to incite trust in their safety-critical system deployments. Effective explanation methods have been thoroughly explored for Convolutional Neural Networks (CNNs), but are still in their infancy for the more prominent Vision Transformers (ViTs). The most popular attribution methods create human-readable explanations using a model's internal information such as class activation maps (Selvaraju et al., 2017), gradients (Simonyan et al., 2014), attention (Vaswani et al., 2017), or relevance scores (Binder et al., 2016). A less abstract method of creating explanations is through feature perturbation. Here, model internals are ignored and input features are masked to discover their effect on the model's decision. However, this class of methods is prohibitively slow for creating the per-pixel explanations required for CNNs (Fisher et al., 2019; Zeiler & Fergus, 2014). Thus, only recently have they seen a resurgence for ViTs (Xie et al., 2023; Englebert et al., 2023) whose reasoning over coarse-grained patches allows the runtime complexity to be controlled. The broad diversity of attribution algorithms necessitates quantitative comparison using metrics for accurate quality assessment.

The most common attribution quality metrics are perturbation-based tests (Petsiuk et al., 2018; Kapishnikov et al., 2019; Walker et al., 2024) which do not need ground-truth labels. The most commonly used perturbation-based tests are insertion and deletion (Petsiuk et al., 2018). These metrics ablate the model input in order of highest value attribution features to determine if the features are as important for the model's decision as indicated. These tests were extended into magnitude aligned scoring (MAS) which also assess the magnitude of the attribution assigned to each feature (Walker et al., 2024). A feature's magnitude is evaluated by measuring the alignment with the

feature's model importance. This raises the question: can we avoid abstract explanation approaches and use these perturbation metrics directly to create better explanations?

In this paper, we propose a method for creating Metric-Driven Attributions for the ViT called (MDA). The approach creates high-quality explanations for ViTs using a perturbation-based approach that mimics the methodology of the attribution quality metrics used to assess the specified attributions. The four main contributions of MDA can be summarized, as follows:

1. We provide the first attribution algorithm that is directly driven by the metrics used to assess the attributions produced by the algorithm. This is a stark departure from existing approaches that mainly leverage internal model representations and methods.

2. The MDA method creates attributions using patch ordering optimization and patch magnitude optimization. The first step orders the patches in terms of importance while the second step assigns an attribution magnitude to each patch while preserving the patch order.

3. Existing metrics encourage sparse explanations that focus on a few important features within an image. We demonstrate that MDA is capable of creating dense attributions that indicate all features of importance by slightly modifying the optimization objective.

4. MDA achieves improved quantitative and qualitative performance compared to 7 state-of-the-art ViT attribution methods. On ImageNet, MDA outperforms the SOTA not only for the perturbation metrics it was optimized for, but also six additional attribution metrics across three ViT models with an average $12\%$ improvement across all metrics.

The remainder of the paper is organized as follows: the related work is in Section 2, the methodology is in Section 3, the experiments are in Section 4, and the conclusion is in Section 5.

## 2 RELATED WORK

In this section, we first present a detailed explanation of the scoring objectives of the perturbation quality metrics of interest. We then discuss the existing approaches to ViT explanation.

### 2.1 PERTURBATION-BASED ATTRIBUTION QUALITY METRICS

Attribution metrics fit into two groups: comparison against ground-truth masks (Borji et al., 2013) or perturbation of the model (Adebayo et al., 2018; Hooker et al., 2019) or the input (Petsiuk et al., 2018; Walker et al., 2024) without ground-truth. Model perturbation randomizes or retrains the entire model to evaluate at scale, which is a slow process. Input perturbation is therefore the most popular because it can evaluate single images without ground-truth or model changes. The insertion and deletion perturbation metrics expect the modification of an important feature in the model input to cause a change in the output (Petsiuk et al., 2018). They measure how well an attribution indicates important features by performing a linear perturbation process between a baseline and an input. We explain insertion with the following text and Figure 1. Given an attribution $A$, the input $X$ of size $D^2$ belonging to class $c$, and the model $F$, the test evaluates $A$ over $N$ steps. Let $P_k$ denote the perturbed image at step $k$. For insertion, $P_0$ denotes the blurred input baseline and $P_N$ is the original input. In each step, the set of $\frac{D^2}{N}$ $i$-th highest magnitude features $A_i \notin P_k$ are selected, seen in Figure 1 (a). These features correspond to the $i$th most important pixels $X_i \notin P_k$, and we update $P_k$ as $P_{k+1} = P_k + X_i$, removing the blurring as seen in Figure 1 (b). The model response is defined as:

$$MR_k = \text{softmax}(F(P_k))_c. \tag{1}$$

Over the $N$ steps of this insertion (ins) perturbation test, the monotonically non-decreasing $MR^{ins}$ curve will be formed on the range $[0, 1]$ as seen in Figure 1 (c). The score of an attribution is then calculated as the area under the $MR$ curve (AUC):

$$score = \frac{1}{N+1} \sum_{k=0}^{N} MR_k. \tag{2}$$

A higher $MR^{ins}$ AUC indicates a better attribution. The AUC in Figure 1(c) is $0.896$. For deletion (del), $P_0$ denotes the original image baseline and $P_N$ is a black image. The $MR^{del}$ curve formed from the test is monotonically non-increasing and has range $[1, 0]$. A lower $MR^{del}$ AUC is better.

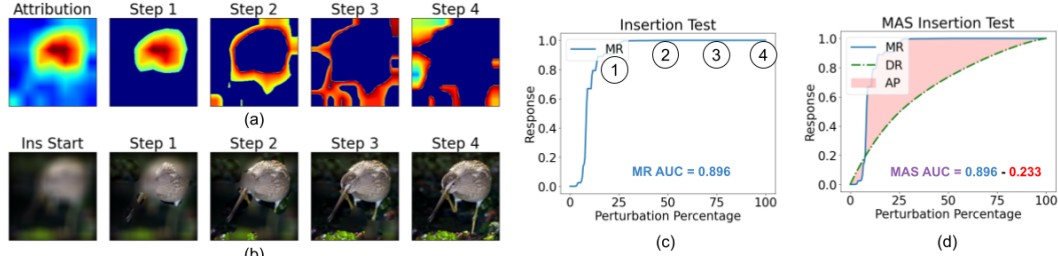

Figure 1: An illustration of the insertion metric where an attribution is evaluated over 4 steps. In each step, the highest attribution values are chosen (a) and those pixels are unblurred (b). The graph (c) illustrates the $MR$ generated by this process as described in Eq (1) and the AUC score from Eq (2). The graph in (d) shows the $MR$, $DR$, and $AP$ as described in Eq (3) and the MAS AUC: $MR - AP$.

The MAS insertion and deletion tests calculate the $MR$ in the same manner (Walker et al., 2024). These tests additionally introduce the density response ($DR$) curve which measures the magnitude of each feature. The $DR$ and $MR$ are compared to determine if high and low value attributions are assigned to important and unimportant features, respectively. In insertion, $DR$ tracks what percentage of the total attribution of $A$ has been inserted at step $k$, and is defined as:

$$DR_k^{ins} = \frac{\sum_{i=0}^{k} |A_i|}{\sum_{i=0}^{N} |A_i|},$$ (3)

where $|.|$ denotes absolute value. $DR^{ins}$ is monotonically increasing and on the range $[0, 1]$, seen as the dashed line in Figure 1 (d). The difference $|MR - DR|$ is called the alignment penalty ($AP$), seen in Figure 1 (d). The MAS score is then $(MR^{ins} - AP)$ AUC, seen as $0.896 - 0.223$ in Figure 1 (d). The $AP$ indicates important (unimportant) features were given too low (high) of a magnitude in the attribution. In deletion, pixels are removed each step, so $DR_k^{del} = (1 - DR_k^{ins})$, the $DR^{del}$ curve is monotonically decreasing and $(MR^{del} + AP)$ AUC is the score.

## 2.2 VISION TRANSFORMER EXPLANATIONS

The ViT model (Dosovitskiy et al., 2020) embeds the input image as a linear array of patch tokens prepended with a classification token [CLS]. These are passed through a decoder-only transformer (Vaswani et al., 2017) which extracts important relationships between all tokens via self-attention, and a final linear layer classifies the output embedding of the [CLS] token. A ViT explanation assigns one value to each of the $M^2$ patches. For ViT $16 \times 16$, the $224 \times 224$px input is broken into $14^2$ patches which are each $16 \times 16$px. The resulting $14^2$ grid of attribution values is upscaled to $224 \times 224$px.

ViT explanations can be derived from the self-attention values of the last layer's [CLS] token (Vaswani et al., 2017), or the matrix multiplication of the self-attention through all layers (Rollout) (Abnar & Zuidema, 2020). However, attention does not contain class information and is a poor explanation alone (Bastings & Filippova, 2020). GradCAM (GC) multiplies the last layer self-attention by its gradients (Selvaraju et al., 2017). Integrated Gradients (IG) generates an explanation from the mean attention gradients of interpolated inputs between a baseline and input Sundararajan et al. (2017). Current SOTA methods mix these techniques. Transformer Attribution (T-Attr) (Chefer et al., 2021a) gathers the attention of all layers with layer-wise relevance propagation (Binder et al., 2016) and multiplies by the gradients. Transition Attention (T-Attn) (Yuan et al., 2021) multiplies Rollout by IG. Bidirectional Attention (Bi-Attn) (Chen et al., 2023) introduces head importance to Rollout and multiplies by IG. Attributions generated by internal model representations are valuable, but most methods sacrifice their theoretical foundations by applying ad-hoc post-processing in this manner.

Due to the ViT patch structure, two perturbation methods show promise without using internal model representations. ViT-CX (Xie et al., 2023) creates a set of masked inputs from the model's output embeddings. These are scored by the model's softmax output and the weighted sum of all masks is the explanation. Transformer Input Sampling (TIS) (Englebert et al., 2023) performs the same process, but masking occurs inside the model at the token embedding level instead of the input level.

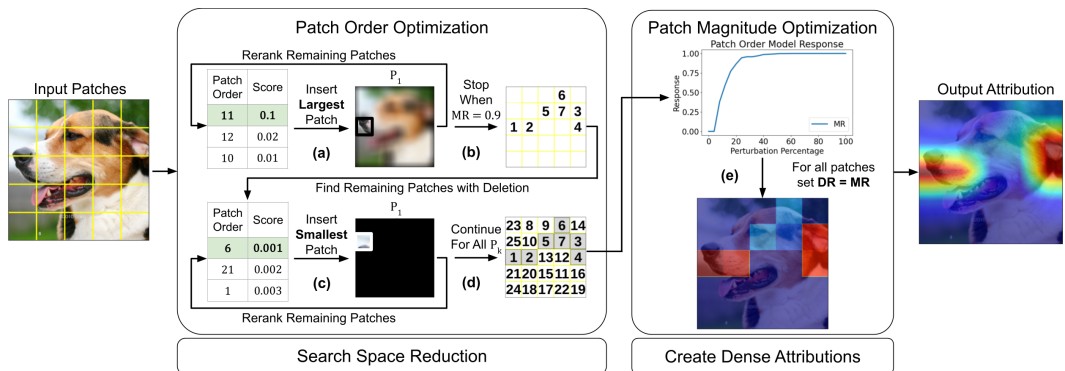

Figure 2: An overview of the MDA method. First, the insertion process finds the patch $(X_i)$ which induces the highest $MR(P_{k+1})$ at each step $k$ (a) until the $MR$ reaches $\tau = 0.9$ (b). This produces the minimum ordering needed for good insertion performance. The remaining patches are found through deletion, where the patch which induces the lowest $MR$ is selected each step (c). This yields a single, optimized ordering of the patches (d). Next, the produced $MR$ informs the magnitude of each patch to be set such that $MR = DR$ (e) (see Figure 3 for details). This yields a sparse, high-scoring attribution. The final attribution is upscaled to the image size.

None of these existing approaches use the attribution quality metrics to generate attributions. By leveraging the metrics' ability to measure attribution quality, we propose to generate attributions that optimize the metric scores, creating high-quality attributions from the source of their evaluation.

## 3 METHODOLOGY

We propose MDA, the first attribution method which generates explanations by directly maximizing attribution quality metric scores. The framework consists of four components: two core optimization steps which generate an attribution and two add-on steps for calibrating and speeding up the process. First, we order the patches to maximize insertion and deletion scores. Second, we set the magnitude of each patch to maximize the MAS insertion and deletion scores. The initial two steps produce sparse attributions with high qualitative scores. Third, we introduce a sliding parameter to allow the creation of dense attributions at the expense of a lower score. Fourth, we present techniques to control the runtime complexity. An overview of the framework is illustrated in Figure 2.

### 3.1 PATCH ORDER OPTIMIZATION

In this section, we describe our patch order optimization technique that orders the $M^2$ patches of the ViT in terms of model importance, which is illustrated in (a) through (d) of Figure 2. The objective of the optimization is to determine the patch order $\mathcal{I}$ which maximizes the insertion and deletion scores. Recall from Eq (2) that the insertion and deletion scores are only a function of the patch order. A potential challenge is that the ideal patch order for insertion may be conflicting with the ideal patch order for deletion. Therefore, our approach attempts to determine a patch order which maximizes the joint insertion and deletion scores. We observe insertion is only sensitive to the order of the first few patches whereas deletion is sensitive to the order of all patches. We utilize this by first determining the patches needed for a strong insertion score, and then find the remaining patches using deletion. We now describe the method for optimizing the insertion and deletion patch orders.

**Insertion-Based Patch Ordering:** Starting with the blurred input image $P_0$, the MDA framework finds the patch $X_i$ that produces the strongest model response and inserts it into the patch order $\mathcal{I}$, which is illustrated in Figure 2(a). The process of iteratively inserting a patch mimics the method for computing the insertion score metric and is therefore expected to score well in terms of insertion. Formally, the patch $X_i$ is found directly from the model response as follows:

$$\underset{i \notin \mathcal{I}}{\arg\max}\, MR(P_k + X_i) \tag{4}$$

where $i$ is the index of any patch $X_i$ not already inserted in $P_k$. Next, $P_{k+1}$ is set to $P_k + X_i$. This process of inserting patches $X_i$ and updating $P_k$ is repeated until $C$ patches have been inserted, where

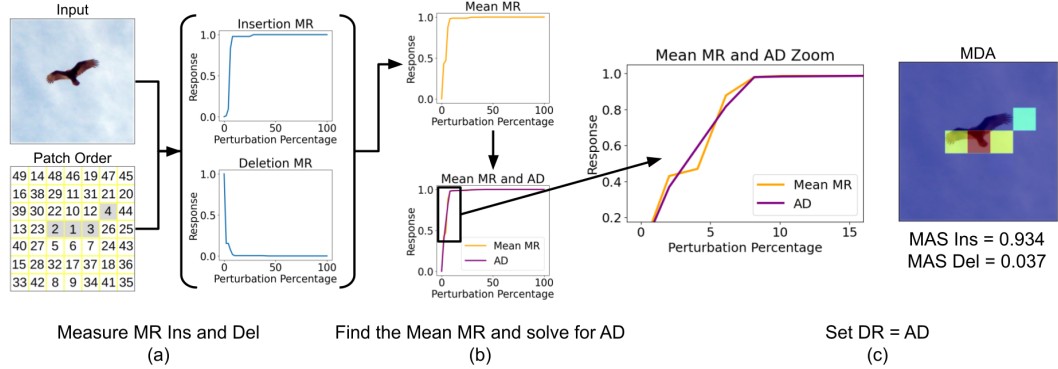

Figure 3: Given the original image and the patch order, we find $MR^{ins}$ and $MR^{del}$ (a). $MR^{del}$ is transformed and averaged with $MR^{ins}$ to find $MR^{mean}$ and a quadratic solver finds the new curve $AD$ from $MR^{mean}$ (b). Note in the zoom, it can be observed that $AD$ is monotonically increasing and has a strictly monotonically decreasing derivative. The $AD$ curve is used to assign patch magnitudes, producing a high MAS score, attribution (c).

$MR(P_C) \geq \tau \cdot MR(P_N)$ with $N = M^2$. In practice, we set the parameter $\tau$ to 0.90. The patches determined by the insertion-based ordering process are shown in Figure 2(b). Next, the remaining $M^2 - C$ patches are determined using the deletion-based patch ordering process.

**Deletion-Based Patch Ordering:** Now, we address the deletion test which transitions from the original image with a high model response to a black image with a low model response. Ideally, each deletion generates a decrease in the model response and patches which contain the most important class information will evoke the smallest model responses (a large decrease in model response). Intuitively, it may seem obvious to find the patch which creates the largest decrease in the model response at every step. However, in deletion, *all important features must be removed* to have a significant decrease in the model response. One ablated patch on an important feature may not remove enough information to change the model response. We therefore reverse the deletion problem to be a modified insertion problem and aim to *minimize* the insertion score.

We begin with a black, low model response image and move towards the original, high model response image. In every perturbation step, we find the patch that has the lowest model response:

$$\underset{i \notin \mathcal{I}}{\arg\min}\, MR(P_k + X_i). \tag{5}$$

This produces a patch order sorted from least to most important, which when reversed, yields the ideal deletion order. As seen in Figure 2, the $C$ patches found from insertion are an input to deletion and locked as the $C$ most important patches. Figure 2 (c) shows the least important patches are modified first, and the final patch order (best deletion order), is found in Figure 2 (d).

## 3.2 PATCH MAGNITUDE OPTIMIZATION

The patch order $\mathcal{I}$ from the previous optimization step maximizes the insertion and deletion scores. Therefore, the objective of the patch magnitude optimization is to minimize the MAS alignment penalty while preserving the patch order $\mathcal{I}$, thus maximizing the MAS score. Let $\mathcal{I}_i$ and $A_i$ denote the ordering index and attribution magnitude of a patch $i$, respectively. We wish to assign a magnitude to each patch such that the order $\mathcal{I}$ is respected as follows:

$$A_i > A_j \quad iff \quad \mathcal{I}_i < \mathcal{I}_j, \quad \forall i \neq j. \tag{6}$$

The patch order $\mathcal{I}$ inherently defines the curves $MR^{ins}$ and $MR^{del}$, seen in Figure 3 (a). To minimize the alignment penalty $AP$, the density response $DR$ should be set to be equal to both the model response $MR^{ins}$ and $MR^{del}$, which is impossible if $MR^{ins}$ and $MR^{del}$ are not identical. Therefore, we first combine $MR^{ins}$ and $MR^{del}$ into $MR^{mean}$ as:

$$MR^{mean} = \frac{(MR^{ins} + (1 - MR^{del}))}{2}, \tag{7}$$

where $(1 - MR^{del})$ transforms $MR^{del}$ to increase like $MR^{ins}$, as seen in Figure 3 (b). Now, we can perform $DR = MR^{mean}$ by setting the attribution value of each patch $A_i$ as follows:

$$A_i = MR_i^{mean} - MR_{i-1}^{mean}, \tag{8}$$

where $A_1 = MR_1^{mean} - MR_0^{mean}$ and $MR_0^{mean} = 0$. However, the assignment of the attributions using Eq (8) does not ensure that the initial patch order would be preserved, i.e., the constraint on the relative ordering of the patches in Eq (6) may be violated, which would in turn change the model response and insertion and deletion scores. For example, if the patch ordering $\mathcal{I} = \{1, 2, 3\}$ generated the model response $MR^{mean} = [0, 3, 3, 4]$, then Eq (8) yields $A_1 = 3$, $A_2 = 0$, and $A_3 = 1$. These assigned attributions result in a revised patch order $\tilde{\mathcal{I}} = \{1, 3, 2\}$ which is inconsistent with the initial ordering of the patches. We explain this behavior in the following paragraph.

To determine the attributions using Eq (8), we specify a new attribution definition $(AD)$ curve which is similar to $MR^{mean}$ and minimizes the alignment penalty. However, we require the curve to satisfy two additional properties which ensure the patch order is preserved, i.e., Eq (6) is satisfied.

1. **Property 1:** The curve $AD$ must be monotonically increasing.
2. **Property 2:** The derivatives of the curve $AD$ must be strictly monotonically decreasing.

The properties are from two observations. 1) All attributions $A_i$ are positive, so $AD$ is, by definition, monotonically increasing. 2) From Eq (8), it can be understood that $AD'_i = A_i$; therefore, to preserve the order for all $A_i$, it is implied that $AD'$ must be strictly monotonically decreasing.

We formulate the problem of specifying the curve $AD$, as follows:

$$\text{Minimize} \quad \frac{1}{N+1} \sum_{i=0}^{N} (AD_i - MR_i^{mean})^2$$
$$\text{subject to} \quad \begin{cases} AD'_{i+1} \leq AD'_i, & \forall i \in [0, N] \\ AD_i \in [0, 1], & \forall i \in [0, N] \\ AD_0 = 0 \text{ and } AD_N = 1 \end{cases}. \tag{9}$$

The objective is to minimize the quadratic distance between $AD$ and $MR^{mean}$ under the linear derivative constraint. This is clearly a quadratic programming problem and a quadratic solver is used to find the solution. Two additional constraints are placed on the range of the function and the endpoints to ensure the final result does not escape the original model response definition.

One final problem remains after finding $AD$. Due to the $\leq$ limitation of the quadratic programming problem, $AD$ has a monotonically non-increasing derivative, so Eq (6) is not satisfied. For example, $AD = [0, 2, 4]$ gives $A_1 = 2$ and $A_2 = 2$, implying $\mathcal{I}_1 = \mathcal{I}_2$ which is impossible. We introduce a patch order term to Eq (8) to induce the monotonically decreasing derivative property:

$$A_i = \Delta AD_i + (\Delta AD_i * \frac{N-i}{N}), \tag{10}$$

where $\Delta AD_i = AD_i - AD_{i-1}$. The additional term ensures $A_i > A_{i+1}$ where $AD'' = 0$. For large $N$, this second term does not increase the $AP$ significantly. We visualize the magnitude optimization in Figure 3 (a) to (c). The final $M \times M$ attribution is upscaled to the input image size.

### 3.3 MAGNITUDE OPTIMIZATION FOR DENSE ATTRIBUTIONS

The MAS metric for evaluating attributions encourages the creation of sparse attributions. We call a set of attributions sparse if they indicate the minimum amount of information needed for accurate model prediction. On the other side of the spectrum, dense attributions indicate all features of an object which are relevant to accurate prediction. We provide an *optional modification* to MDA which allows a user to specify if the generated attributions are sparse, dense, or a mixture of the two.

Given $A^{sparse}$ as defined from Eq (10), we find the dense attribution $A^{dense}$ by assigning all important patches a magnitude which represents their order:

$$A_i^{dense} = \begin{cases} \frac{N-i}{N} & \text{if } A_i^{sparse} \geq \kappa \\ A_i^{sparse} & \text{if } A_i^{sparse} < \kappa \end{cases}. \tag{11}$$

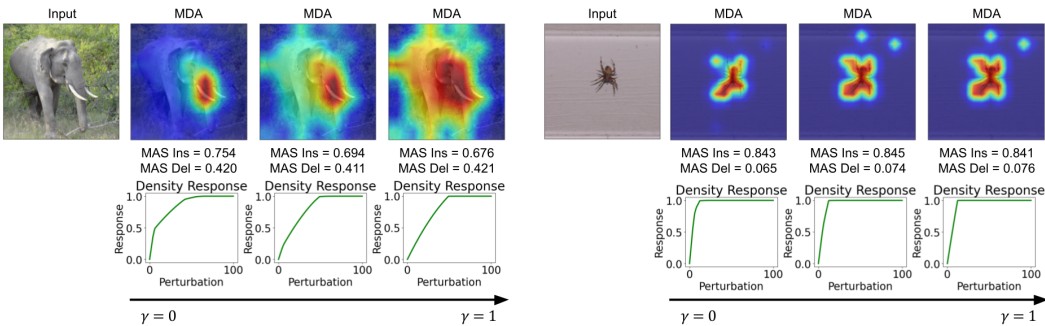

Figure 4: We illustrate MDA for the transition of $\gamma = 0$ to $\gamma = 1$. For the elephant, the attribution grows as the magnitude is more evenly spread among features (seen by the more linear $DR$), indicating only the tusks are needed for classification. For the spider, the $DR$ and attribution do not greatly change, indicating all features of the spider are important for classification.

The $\kappa$ parameter controls the model importance cutoff since $A_i \approx \Delta MR_i$. In practice, we employ $\kappa = 0.005$ to only strongly attribute patches with more than $0.5\%$ model importance.

We now introduce a user-controlled parameter $\gamma$ to blend between $A^{sparse}$ and $A^{dense}$ where $\gamma = 0$ and $\gamma = 1$ yield sparse and dense attributions, respectively:

$$A^{\gamma} = (1 - \gamma)A^{sparse} + \gamma A^{dense}. \tag{12}$$

In Figure 4 we illustrate this for a large and small subject. We see for the large subject, the attribution expands to cover all features, resulting in a more linear density response. For the small subject, the attribution does not expand, indicating all important features were already used for classification. A user can tune $\gamma$ to their choosing, but, quantitatively, the best explanation is created with $\gamma = 0$.

## 3.4 SEARCH SPACE REDUCTION

In this section we provide solutions to improving the runtime of MDA. A naive implementation of the optimization search over the $M^2$ coarse-grained ViT patches results in a $\mathcal{O}(M^4)$ runtime. This optimization problem resembles the knapsack problem, but we cannot reduce the search space through common techniques such as memoization or tabulation. In this problem, unlike knapsack, the next optimal value $MR(P_{k+1})$ is unknown and cannot be assumed to be additive, i.e. $MR(P_1 + X_i) \neq MR(P_1) + MR(P_0 + X_i)$ where each $X_i$ was previously enumerated in the search for $P_1$.

As typical techniques will not work, we take heuristic approaches to reduce the search space, approaching the absolute minimum $\mathcal{O}(M^2)$ runtime. First, the insertion optimization phase is improved. As we terminate the insertion optimization when we find the $C$ patches needed to produce an $MR = 0.9$ we reduce the search space to $\mathcal{O}(CM^2)$. Across 100 images, only 48/196 and 6.5/49 patches were needed to reach this model response for the ViT-base $16 \times 16$ and ViT-base $32 \times 32$ models, respectively, resulting in $\mathcal{O}(M^3)$ insertion search space. For deletion, the order of all patches is needed, so an early stopping technique cannot be employed, yielding a runtime of $\mathcal{O}(M^4)$. However, we can prune the search space of both insertion and deletion.

We can take advantage of the existing ViT attribution method space, and use a SOTA explanation to provide an initial ordering of the patches in the image. We choose Bi-Attn (Chen et al., 2023) due to its low computational cost and strong performance among other ViT methods. For insertion, the input attribution provides an ordering of all patches from high to low magnitude. We use this ordering to inform a smaller search space, i.e. instead of searching through all $M^2 - k$ patches not inserted in each step $k$, we search through the first $2M$ patches in the ordering which we can assume are valuable to the model. For deletion, the input attribution provides an ordering from lowest to highest value and we search through the first $2M$ patches in the ordering which we can assume are not valuable to the model. These two steps combined reduce the runtime from $\mathcal{O}(M^4)$ to $\mathcal{O}(M^2) + \mathcal{O}(M^3) = \mathcal{O}(M^3)$ and finally, we can approach $\mathcal{O}(M^2)$ with batched execution. The runtime could be reduced to the minimum $\mathcal{O}(M^2)$ by selecting $M$ patches at every insertion step instead of 1, but this requires better attributions than what exists to avoid a significant quality penalty. We now evaluate the proposed MDA method with $\gamma = 0$ and Bi-Attn (Chen et al., 2023) as input.

## 4 EXPERIMENTAL EVALUATION

In this section we present qualitative and quantitative evaluations of the proposed MDA ViT attribution method. We perform all experiments using PyTorch (Paszke et al., 2019) and use the ImageNet 2012 validation dataset (Russakovsky et al., 2015) and the ImageNet Segmentation dataset (Guillaumin et al., 2014). We employ three ViT models: ViT-base $16 \times 16$, ViT-tiny $16 \times 16$, and ViT-base $32 \times 32$ as defined in the ViT paper (Dosovitskiy et al., 2020). These models are trained on ImageNet and are from the PyTorch Image Models repository (Wightman, 2019). The experiments were run on one server with four NVIDIA A40 GPUs. We compare our method against GC (Selvaraju et al., 2017), IG (Sundararajan et al., 2017), T-Attn (Yuan et al., 2021), T-Attr (Chefer et al., 2021a), ViT-CX (Xie et al., 2023), TIS (Englebert et al., 2023), and Bi-Attn (Chen et al., 2023).

### 4.1 QUANTITATIVE EVALUATION

In this section, we provide quantitative evaluation of our MDA method's performance, and evaluate the parameters used in MDA. We will first evaluate how MDA performs in the perturbation tests that we have have optimized the framework for (Petsiuk et al., 2018; Walker et al., 2024). Next, we evaluate how these optimized attributions perform for three other metrics: pointing game (Chefer et al., 2021a), positive and negative perturbation (Chefer et al., 2021a; DeYoung et al., 2020), and monotonicity (Arya et al., 2019). Following these, we evaluate how MDA is affected by the selection of hyperparameters: $\tau$, $\gamma$, and $\kappa$. Lastly, we evaluate runtime. Due to space constraints and their lower importance, we defer evaluations of monotonicity, $\tau$, $\gamma$, and $\kappa$ to the Appendix.

**Optimized Perturbation Metric Evaluation:** We have thoroughly described insertion, deletion, and the MAS variants in previous sections. For these tests, all input images are $224 \times 224$px and we use a step size of 224px, for a total of 224 perturbation steps as in the original implementations. Additionally, for insertion, we use a blurring kernel which results in a softmax score of less than $1\%$, to improve test accuracy. We use the code provided by the repositories for insertion/deletion and MAS (Petsiuk et al., 2018; Walker et al., 2024). The results in Table 1 compare all 8 attribution methods over 5000 ImageNet images with 5 images per class for the perturbation metrics. We see significant improvements by MDA in all tests, except for deletion, where it faces a loss. We see a minimum improvement of $9\%$ in insertion - deletion, and a maximum improvement of $51\%$ in MAS insertion - deletion. The deletion performance of MDA is easily explained by the definition of our patch order optimization. As we optimize the first few patches for a high insertion score, deletion can be penalized as a result (we explore this in A.6). However, as shown by insertion - deletion scores, joint optimization nets a positive improvement over all attribution methods. We show the results of this test on ViT-base $32 \times 32$ and ViT-tiny $16 \times 16$ models in Tables 4 and 5 in the Appendix.

**Pointing Game Evaluation:** In the pointing game, we measure the intersection-over-union (IoU), mean average precision (MAP), and F1 scores of each method's pointing accuracy when compared to ground-truth segmentation maps of ImageNet Segmentation data. This test uses the default code available in the original author's repository (Chefer et al., 2021b). For each attribution method, a threshold is applied, binarizing the attribution to make pointing game accuracy measurements. The metric authors use an attribution's mean as the threshold. However, since this threshold is equivalent to $\gamma$ in MDA we select the best $\gamma$ for MDA given the three output scores. The results are shown in Table 2 for the ViT-base $32 \times 32$ model. First, we see that choosing the ideal $\gamma$ for MDA to perform

Table 1: Evaluation of MDA on the optimized metrics for the ViT-base $16 \times 16$ model.

| Test Type | Metric From Petsiuk et al. (2018) | | | Metric From Walker et al. (2024) | | |
| --- | --- | --- | --- | --- | --- | --- |
| | **Ins ($\uparrow$)** | **Del ($\downarrow$)** | **Ins - Del ($\uparrow$)** | **Ins ($\uparrow$)** | **Del ($\downarrow$)** | **Ins - Del ($\uparrow$)** |
| GC (Selvaraju et al., 2017) | 0.737 | 0.241 | 0.496 | 0.622 | 0.312 | 0.311 |
| IG (Sundararajan et al., 2017) | 0.741 | 0.222 | 0.518 | 0.623 | 0.349 | 0.274 |
| ViT-CX (Xie et al., 2023) | 0.722 | 0.236 | 0.486 | 0.582 | 0.408 | 0.174 |
| T-Attn (Yuan et al., 2021) | 0.748 | 0.228 | 0.520 | 0.631 | 0.347 | 0.284 |
| T-Attr (Chefer et al., 2021a) | 0.741 | 0.232 | 0.508 | 0.638 | 0.331 | 0.307 |
| Bi-Attn (Chen et al., 2023) | 0.760 | 0.218 | 0.542 | 0.649 | 0.320 | 0.329 |
| TIS (Englebert et al., 2023) | 0.761 | **0.196** | 0.565 | 0.615 | 0.385 | 0.230 |
| MDA (ours) | **0.856** | 0.232 | **0.624** | **0.775** | **0.279** | **0.497** |

Table 2: ImageNet segmentation results for the ViT-base $32 \times 32$ model.

| Metric | MAP | IoU | F1 |
|---|---|---|---|
| GC (Selvaraju et al., 2017) | 0.715 | 0.570 | 0.366 |
| IG (Sundararajan et al., 2017) | 0.687 | 0.573 | 0.462 |
| ViT-CX (Xie et al., 2023) | 0.678 | 0.535 | 0.428 |
| T-Attn (Yuan et al., 2021) | 0.705 | 0.594 | 0.465 |
| T-Attr (Chefer et al., 2021a) | 0.758 | 0.648 | 0.459 |
| Bi-Attn (Chen et al., 2023) | 0.753 | 0.653 | 0.491 |
| TIS (Englebert et al., 2023) | 0.705 | 0.586 | 0.460 |
| MDA (ours) | **0.796** | **0.702** | 0.487 |
| MDA F1 Focused (ours) | 0.760 | 0.661 | **0.504** |

well on all three tests leads to a win for MAP and IoU, but a minor loss for F1. However, if we choose $\gamma$ such that F1 score is prioritized, we achieve an F1 win while retaining a win in MAP and IoU. These results show that optimizing MDA for insertion and deletion extends to high performance in other metrics, making MDA more valuable.

**Positive and Negative Perturbation Evaluation:** The positive and negative perturbation tests functional similarly to the optimized metrics, but measure prediction accuracy instead of softmax score. They measure an attribution's ability to explain the model with a two-stage process. First, for a given set of validation images, an attribution is made for all images. Following this, the pixels of the images are gradually masked, and the resulting accuracy is recorded at each step, forming a curve. In positive perturbation, pixels are masked in largest attribution order, and in negative perturbation they are masked in lowest attribution order, thus lower and higher AUC scores are better, respectively. We perform these tests with the default code available in the original author's repository (Chefer et al., 2021b) using the ImageNet validation set (Russakovsky et al., 2015) on the ViT-base $32 \times 32$ model and we show the results in Table 3. In this table we clearly see MDA outperforms the SOTA methods in both tests. These indicate not only that MDA best orders the most important features but it also best orders the least important features. Since this test measures via model accuracy and not softmax score, we can confirm MDA was not optimized for this test, yet still outperforms all methods.

Table 3: Positive and negative perturbation tests for the ViT-base $32 \times 32$ model.

| Metric | Positive ($\downarrow$) | Negative ($\uparrow$) |
|---|---|---|
| GC (Selvaraju et al., 2017) | 0.199 | 0.637 |
| IG (Sundararajan et al., 2017) | 0.128 | 0.698 |
| ViT-CX (Xie et al., 2023) | 0.143 | 0.708 |
| T-Attn (Yuan et al., 2021) | 0.140 | 0.714 |
| T-Attr (Chefer et al., 2021a) | 0.140 | 0.708 |
| Bi-Attn (Chen et al., 2023) | 0.124 | 0.733 |
| TIS (Englebert et al., 2023) | 0.126 | 0.741 |
| MDA (ours) | **0.122** | **0.789** |

**Runtime Evaluation:** We measure the mean runtime of MDA to be 13.34s, 3.64s, and 1.13s, for the ViT-base $16 \times 16$, ViT-tiny $16 \times 16$, and ViT-base $32 \times 32$ models, respectively. We provide a comparison of these times with SOTA attributions in the Appendix and provide further discussion about the value of online and offline attribution methods.

### 4.2 QUALITATIVE EVALUATION

In this section we perform qualitative MDA $\gamma$ selection and compare MDA against the SOTA methods. In the Appendix, we extended these evaluations and explore seed attribution selection.

**Evaluation of $\gamma$:** We compare MDA with $\gamma = 0$ and $\gamma = 1$ on the ViT-base $32 \times 32$ model against Bi-Attn in Figure 5 and we make three distinctions. First, existing SOTA methods suffer from attribution on unimportant features, hurting quantitative and qualitative value. Second, at a low $\gamma$, MDA provides minimal information attributions, indicating only the most important features. Third,

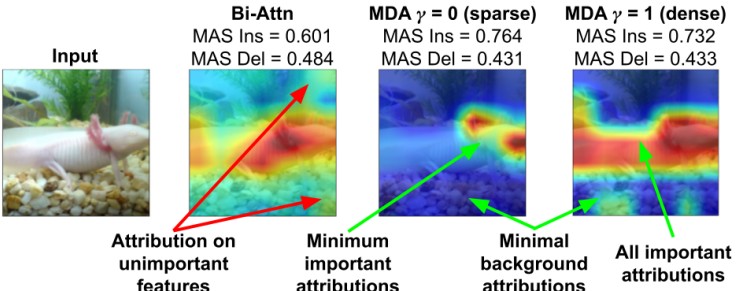

Figure 5: MDA has major visual improvements over the state-of-the-art methods. Unlike Bi-Attn (Chen et al., 2023), MDA does not provide attributions for any features unrelated to the class subject. In addition, it can provide a sparse or dense attribution while scoring more favorably.

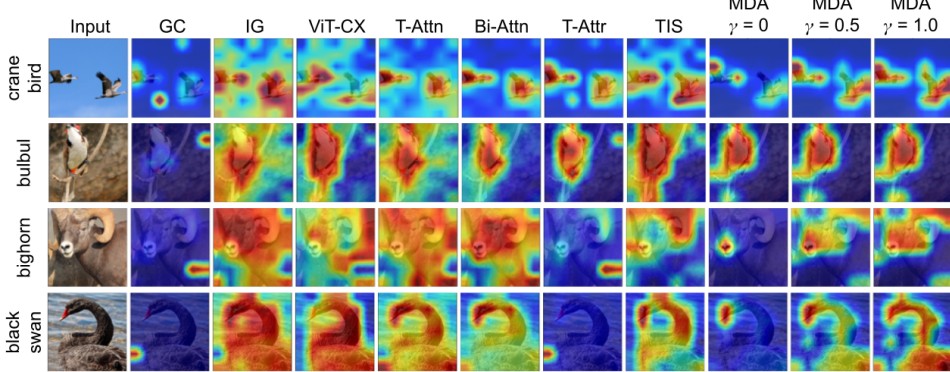

Figure 6: In the selected ImageNet examples we compare MDA with $\gamma = 0$, $\gamma = 0.5$, and $\gamma = 1$ against GC (Selvaraju et al., 2017), IG (Sundararajan et al., 2017), T-Attn (Yuan et al., 2021), T-Attr (Chefer et al., 2021a), ViT-CX (Xie et al., 2023), TIS (Englebert et al., 2023), and Bi-Attn (Chen et al., 2023). We see in all examples, MDA $\gamma = 0$ provides the best sparse attribution with no background attributions. As $\gamma$ increases, MDA provides dense attributions which lack unimportant attributions.

MDA at a high $\gamma$ provides dense attributions, highlighting all important features, while avoiding unimportant features. We see this behavior continues in additional examples in the Appendix.

**SOTA Comparison:** We present four qualitative comparison examples from the ImageNet dataset in Figure 6 for the ViT-base $32 \times 32$ model. We show MDA with $\gamma = 0$, $\gamma = 0.5$, and $\gamma = 1$ across a mix of images with a large and small subject. For all examples, $\gamma = 0$ provides a sparse, minimal information attribution which highlights the most important features. As $\gamma$ moves to 1, attributions cover more of the image subject and stay retained to the image subject. In all examples, MDA provides reduced attributions on unimportant features when compared to the existing state-of-the-art ViT attribution methods. We show 255 more examples in Appendix A.10 equally divided among the three models. We see the behavior shown in this figure continues across the images presented, indicating that MDA provides more valuable attributions than state-of-the-art methods.

## 5 CONCLUSION

We propose the first-of-its-kind Metric-Driven Attribution for the vision transformer. Through two core optimization objectives, MDA generates attributions which perform highly in the insertion, deletion, and MAS attribution quality metrics. The first objective finds an ideal patch order which jointly maximizes insertion and deletion scores. The second objective finds the ideal magnitude of each patch to jointly maximize the MAS insertion and deletion scores. The result is a high-scoring, sparse explanation for only the most important features in an image. Through a modification of the objective, MDA can additionally create dense attributions, allowing a user to choose the sparse-dense balance. Across three ViT models, MDA shows significant performance gains over SOTA methods, further motivating optimization for attribution metrics. Due to its design, MDA can be applied to any input perturbation metric. The problem of defining the ideal metric is still open, and MDA can evolve with new metrics to create stronger attributions.

ACKNOWLEDGMENTS

This material is based on research sponsored by DARPA Agreement FA8750-23-2-0501, DOE Awards: DE-SC0023494, DE-SC0024428, and DE-SC0024576, and startup funds from the University of Florida. The U.S. Government is authorized to reproduce and distribute reprints for Governmental purposes notwithstanding any copyright notation thereon. The views and conclusions contained herein are those of the authors and should not be interpreted as necessarily representing the official policies or endorsements, either expressed or implied, of DARPA, DOE, or the U.S. Government.

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

# A APPENDIX

In this section, we present extended quantitative and qualitative results. First, we present the additional tables referenced in the experimental evaluation to present results across all three models. Then, we present detailed and extensive qualitative results across all three models.

## A.1 LIMITATIONS

In this work, we compare MDA against seven SOTA ViT attribution methods. We recognize that a portion of the quantitative tests are performed with perturbation-based attribution quality metrics. While MDA only explicitly optimizes for the RISE and MAS tests, it can be considered an advantage to compare against other XAI methods on any perturbation-based metric. All XAI methods we compare with are implemented via their recommended parameters, but it is conceivable that their performance could be "optimized" via careful tuning of their input parameters. However, we note that MDA does not require any parameter tuning and will always produce an optimized result, avoiding undue burden on the user. Furthermore, MDA shows superior performance over all compared SOTA methods on the ground-truth ImageNet segmentation test which MDA does not optimize for, thus showing it can maintain its position as the new SOTA in an unbiased test.

## A.2 EXTENDED OPTIMIZATION METRIC EVALUATION

We include Tables 4 and 5 which provide the results of the main experimental comparison for the ViT-base $32 \times 32$ and ViT-tiny $16 \times 16$ models. The results seen in these tables are consistent with the Table 1. The only difference is in Table 5 where the winner of deletion is ViT-CX. However, referencing Figure 7, we can still conclude this is due to our joint optimization of insertion and deletion, and MDA would clearly win over ViT-CX in deletion if optimized solely for deletion. In Table 4, we note a maximum improvement of 73% in MAS insertion - deletion.

Table 4: Evaluation of MDA on the optimized metrics for the ViT-base $32 \times 32$ model.

| | Metric From Petsiuk et al. (2018) | | | Metric From Walker et al. (2024) | | |
|---|---|---|---|---|---|---|
| **Test Type** | **Ins (↑)** | **Del (↓)** | **Ins - Del (↑)** | **Ins (↑)** | **Del (↓)** | **Ins - Del (↑)** |
| GC (Selvaraju et al., 2017) | 0.783 | 0.262 | 0.520 | 0.695 | 0.339 | 0.356 |
| IG (Sundararajan et al., 2017) | 0.805 | 0.206 | 0.598 | 0.575 | 0.425 | 0.151 |
| ViT-CX (Xie et al., 2023) | 0.813 | 0.208 | 0.605 | 0.615 | 0.389 | 0.226 |
| T-Attn (Yuan et al., 2021) | 0.797 | 0.222 | 0.576 | 0.578 | 0.427 | 0.151 |
| T-Attr (Chefer et al., 2021a) | 0.798 | 0.221 | 0.576 | 0.650 | 0.358 | 0.292 |
| Bi-Attn (Chen et al., 2023) | 0.826 | 0.205 | 0.621 | 0.613 | 0.396 | 0.216 |
| TIS (Englebert et al., 2023) | 0.847 | **0.176** | 0.671 | 0.643 | 0.370 | 0.273 |
| MDA (ours) | **0.908** | 0.200 | **0.708** | **0.851** | **0.235** | **0.616** |

Table 5: Evaluation of MDA on the optimized metrics for the ViT-tiny $16 \times 16$ model.

| | Metric From Petsiuk et al. (2018) | | | Metric From Walker et al. (2024) | | |
|---|---|---|---|---|---|---|
| **Test Type** | **Ins (↑)** | **Del (↓)** | **Ins - Del (↑)** | **Ins (↑)** | **Del (↓)** | **Ins - Del (↑)** |
| GC (Selvaraju et al., 2017) | 0.729 | 0.252 | 0.477 | 0.623 | 0.323 | 0.299 |
| IG (Sundararajan et al., 2017) | 0.745 | 0.233 | 0.512 | 0.627 | 0.355 | 0.272 |
| ViT-CX (Xie et al., 2023) | 0.719 | **0.155** | 0.563 | 0.548 | 0.387 | 0.161 |
| T-Attn (Yuan et al., 2021) | 0.752 | 0.239 | 0.513 | 0.635 | 0.354 | 0.281 |
| T-Attr (Chefer et al., 2021a) | 0.745 | 0.243 | 0.502 | 0.643 | 0.339 | 0.304 |
| Bi-Attn (Chen et al., 2023) | 0.764 | 0.228 | 0.536 | 0.654 | 0.328 | 0.326 |
| TIS (Englebert et al., 2023) | 0.764 | 0.207 | 0.557 | 0.617 | 0.389 | 0.228 |
| MDA (ours) | **0.858** | 0.241 | **0.617** | **0.777** | **0.289** | **0.488** |

### A.3 Monotonicity Evaluation

The last quantitative attribution metric we include is the monotonicity test (Arya et al., 2019). This test measures if inserting all features in order of largest attribution value will lead to a monotonically increasing curve model confidence. Starting from a blurred image, features are inserted and the confidence of the image class prediction is measured. Over all features, this creates the confidence vector $\mathbf{c}$. This vector is then compared to a feature order vector $\mathbf{o}$ which denotes the descending token importance. Monotonicity is then measured as $MONO = p(\mathbf{o}, \mathbf{c})$, where p(.) is the Spearman correlation of the vectors, and a higher score means $\mathbf{c}$ is more monotonic. We report the results in Table 6 for the ViT-base $32 \times 32$ model using 100 ImageNet validation dataset images. We see MDA significantly improves over the SOTA methods. This further proves that *MDA meets objectives outside of its optimization targets*, showing its value as an effective attribution for ViT models.

Table 6: Monotonicity comparison on the ViT-base $32 \times 32$ model.

| Metric | Monotonicity (↑) |
|---|---|
| GC (Selvaraju et al., 2017) | 0.798 |
| IG (Sundararajan et al., 2017) | 0.791 |
| ViT-CX (Xie et al., 2023) | 0.775 |
| T-Attn (Yuan et al., 2021) | 0.782 |
| T-Attr (Chefer et al., 2021a) | 0.788 |
| Bi-Attn (Chen et al., 2023) | 0.783 |
| TIS (Englebert et al., 2023) | 0.770 |
| MDA (ours) | **0.820** |

### A.4 Quantitative $\tau$ Evaluation

We now explain the deletion loss in Tables 1, 4, and 5, with an ablation study of $\tau$ in Figure 7. In graph (a), the dots represent the insertion and deletion scores of the attributions and in (b), the dots are the MAS scores. Better overall scores are closer to the origin. The 11 dots are generated by varying $\tau$ from 1 to 0 (decreasing by 0.1 left to right). As $\tau$ decreases, the deletion score improves and insertion suffers. When $\tau = 1$, MDA is optimized solely for insertion (only a blurred baseline is used) and when $\tau = 0$, it is optimized for deletion only (only a black baseline is used). This is in contrast to the implementation of $\tau = 0.9$ which leads to a joint insertion and baseline patch ordering, thus both the blurred and black baselines are used, respectively. We circle $\tau = 0.9$ in red and draw a boundary across the MDA dots. We observe no SOTA methods pass the boundary, signifying *there exists a configuration of MDA which can win both tests*. In (c), we show selecting $\tau = 0.9$, on average, provides a peak in the joint scores, indicating it is the correct choice. This analysis was performed on 100 ImageNet images for the ViT-base $32 \times 32$ model.

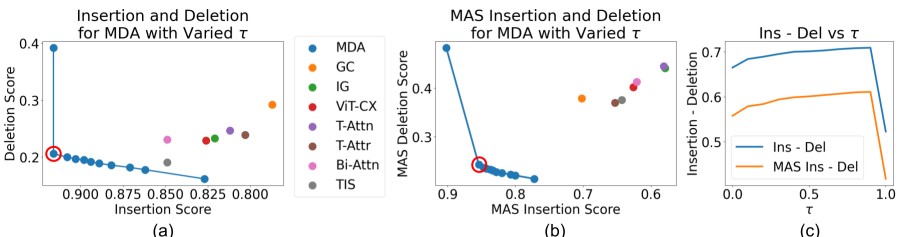

Figure 7: We illustrate the impact of $\tau$ for the insertion, deletion (a) MAS tests (b), and insertion - deletion (c). In (a) and (b), a score near the origin is better and we circle $\tau = 0.9$ in red. We draw a boundary across all MDA points and compare the SOTA attributions. We see no method breaks this boundary, indicating there is always a configuration of MDA which will perform best. We show $\tau = 0.9$ provides the peak score (c), verifying it is the best choice for joint optimization.

We present in Tables 7, 8, and 9, tabular data of this $\tau$ experiment for the ViT-base $16 \times 16$, ViT-tiny $16 \times 16$, and and ViT-base $32 \times 32$ models when only MDA is considered. As expected, we see for all models that MDA optimized for insertion ($\tau = 1$), deletion ($\tau = 0$), or with joint optimization ($\tau = 0.9$) will perform the best on the insertion, deletion, or insertion - deletion tests, respectively.

Table 7: Evaluation of MDA optimization of insertion or deletion on the ViT-base $16 \times 16$ model.

| Test Type | Metric From Petsiuk et al. (2018) | | | Metric From Walker et al. (2024) | | |
|---|---|---|---|---|---|---|
| | Ins (↑) | Del (↓) | Ins - Del (↑) | Ins (↑) | Del (↓) | Ins - Del (↑) |
| MDA Insertion | **0.888** | 0.501 | 0.387 | **0.875** | 0.649 | 0.226 |
| MDA Deletion | 0.710 | **0.151** | 0.559 | 0.581 | **0.216** | 0.364 |
| MDA (ours) | 0.864 | 0.259 | **0.605** | 0.781 | 0.309 | **0.472** |

Table 8: Evaluation of MDA optimization of insertion or deletion on the ViT-tiny $16 \times 16$ model.

| Test Type | Metric From Petsiuk et al. (2018) | | | Metric From Walker et al. (2024) | | |
|---|---|---|---|---|---|---|
| | Ins (↑) | Del (↓) | Ins - Del (↑) | Ins (↑) | Del (↓) | Ins - Del (↑) |
| MDA Insertion | **0.896** | 0.236 | 0.660 | **0.880** | 0.313 | 0.567 |
| MDA Deletion | 0.706 | **0.062** | 0.644 | 0.640 | **0.143** | 0.496 |
| MDA (ours) | 0.882 | 0.140 | **0.742** | 0.829 | 0.163 | **0.666** |

Table 9: Evaluation of MDA optimization of insertion or deletion on the ViT-base $32 \times 32$ model.

| Test Type | Metric From Petsiuk et al. (2018) | | | Metric From Walker et al. (2024) | | |
|---|---|---|---|---|---|---|
| | Ins (↑) | Del (↓) | Ins - Del (↑) | Ins (↑) | Del (↓) | Ins - Del (↑) |
| MDA Insertion | **0.916** | 0.392 | 0.523 | **0.902** | 0.483 | 0.418 |
| MDA Deletion | 0.827 | **0.162** | 0.665 | 0.772 | **0.214** | 0.558 |
| MDA (ours) | 0.916 | 0.206 | **0.709** | 0.854 | 0.243 | **0.611** |

Lastly, in Figure 8, we perform a case study to illustrate that patch order is unique to each combination of optimization and baseline. In these images, the dark red represents the most important patch and dark blue represents the least important. In the first row we compare insertion (a), deletion (b), and joint optimization (c) using the black baseline. We see these combinations produce distinct patch orders as supported by the scores. In the second row: (d), (e), and (f), we compare these optimizations with the blurred baseline and see the same behavior. Finally, in (g) we present the proposed joint optimization which comes from insertion using the blurred baseline (d) and deletion using the black baseline (b). Of note, we see that (b) and (d) have the highest metric scores with respect to their optimization metric due to the right baseline choice, which produces the best score when using joint optimization (g), supporting our design choice of joint optimization.

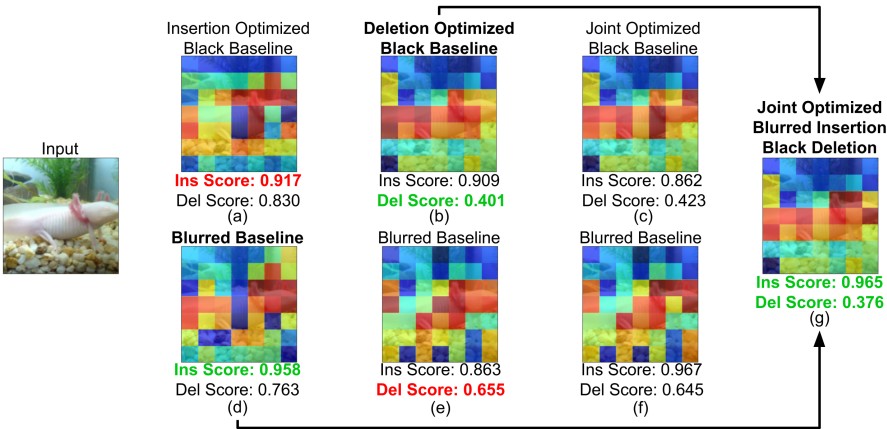

Figure 8: A visual comparison of the *unique* patch orders resulting from the various combinations of insertion or deletion optimization with black and blurred baselines and the resulting joint optimization. The darkest red represents the first patch in the order, and the darkest blue represents the last.

## A.5 RUNTIME EVALUATION

We perform a runtime evaluation of all attribution methods under comparison. We report the mean and standard deviation of a method's runtime in seconds over 100 ImageNet (Russakovsky et al., 2015) images for the ViT-base $16 \times 16$, ViT-tiny $16 \times 16$ and $32 \times 32$ models. These results are shown in Table 10. We see in the results that MDA is clearly slower, but its runtime in the range of 1.1 to 13.3 seconds is still *very reasonable for real-world use*. Attributions are often discussed as being desirable for real-time use. However, we argue that the use of attributions should not be limited to real-time scenarios only. Specifically in the case of MDA, we believe its slower runtime is a fair trade off not only for its performance, but also its flexibility which benefits offline utilization, and its lack of need for internal model information. MDA not only achieves improvements of up to 73% over SOTA methods, but it is the sole attribution method which is flexible to user preferences in providing sparse or dense attributions. This second attribute is valuable for offline evaluation of a neural model, especially as $\gamma$ is applied in post without model calls such that only one MDA generation must occur to have access to a varied set of attributions from $\gamma \in [0, 1]$. The flexibility could assist in debugging in ways that current attribution methods cannot.

Table 10: Runtime comparison on the ViT-b $16 \times 16$, ViT-t $16 \times 16$, and ViT-b $32 \times 32$ models.

| | Mean Runtime per Image (s) | | |
|---|---|---|---|
| Model | ViT-b $16 \times 16$ | ViT-t $16 \times 16$ | ViT-b $32 \times 32$ |
| GC (Selvaraju et al., 2017) | $\mathbf{0.014} \pm 0.006$ | $\mathbf{0.008} \pm 0.005$ | $\mathbf{0.010} \pm 0.006$ |
| IG (Sundararajan et al., 2017) | $0.226 \pm 0.007$ | $0.116 \pm 0.012$ | $0.144 \pm 0.007$ |
| ViT-CX (Xie et al., 2023) | $0.519 \pm 0.078$ | $0.189 \pm 0.020$ | $0.609 \pm 0.058$ |
| T-Attn (Yuan et al., 2021) | $0.225 \pm 0.007$ | $0.122 \pm 0.012$ | $0.143 \pm 0.008$ |
| T-Attr (Chefer et al., 2021a) | $0.049 \pm 0.006$ | $0.048 \pm 0.008$ | $0.049 \pm 0.008$ |
| Bi-Attn (Chen et al., 2023) | $0.226 \pm 0.007$ | $0.118 \pm 0.011$ | $0.141 \pm 0.008$ |
| TIS (Englebert et al., 2023) | $1.104 \pm 0.015$ | $0.171 \pm 0.011$ | $0.359 \pm 0.015$ |
| MDA (ours) | $13.344 \pm 0.308$ | $3.639 \pm 0.649$ | $1.130 \pm 0.249$ |

## A.6 QUANTITATIVE $\gamma$ EVALUATION

We compare MDA with $\gamma = 0.0$, $\gamma = 0.5$, and $\gamma = 1.0$ on 100 ImageNet images. Table 11 shows the ViT-base $16 \times 16$ model results. As expected, MDA $\gamma = 0$ achieves the best scores in all tests. However, the score penalty of increasing $\gamma$ is not extreme, indicating dense attributions are still of value. We provide results for the ViT-tiny $16 \times 16$ and ViT-base $32 \times 32$ models in Tables 12 and 13 below. We find the same results to hold true. We later provide more detailed qualitative examples to compare MDA against SOTA methods and further analyze the effect of $\gamma$ across the models.

Table 11: Evaluation of MDA with varying levels of $\gamma$ on the ViT-base $16 \times 16$ model.

| | Metric From Petsiuk et al. (2018) | | | Metric From Walker et al. (2024) | | |
|---|---|---|---|---|---|---|
| Test Type | Ins ($\uparrow$) | Del ($\downarrow$) | Ins - Del ($\uparrow$) | Ins ($\uparrow$) | Del ($\downarrow$) | Ins - Del ($\uparrow$) |
| MDA $\gamma = 1$ | 0.812 | 0.350 | 0.462 | 0.701 | 0.466 | 0.235 |
| MDA $\gamma = 0.5$ | 0.849 | 0.262 | 0.587 | 0.781 | 0.312 | 0.472 |
| MDA $\gamma = 0$ (ours) | **0.864** | **0.259** | **0.605** | **0.781** | **0.309** | **0.472** |

Table 12: Evaluation of MDA with varying levels of $\gamma$ on the ViT-tiny $16 \times 16$ model.

| | Metric From Petsiuk et al. (2018) | | | Metric From Walker et al. (2024) | | |
|---|---|---|---|---|---|---|
| Test Type | Ins ($\uparrow$) | Del ($\downarrow$) | Ins - Del ($\uparrow$) | Ins ($\uparrow$) | Del ($\downarrow$) | Ins - Del ($\uparrow$) |
| MDA $\gamma = 1$ | 0.861 | 0.159 | 0.703 | 0.792 | 0.194 | 0.599 |
| MDA $\gamma = 0.5$ | 0.882 | 0.140 | 0.742 | 0.827 | 0.163 | 0.665 |
| MDA $\gamma = 0$ (ours) | **0.882** | **0.140** | **0.742** | **0.829** | **0.163** | **0.666** |

Table 13: Evaluation of MDA with varying levels of $\gamma$ on the ViT-base $32 \times 32$ model.

| | Metric From Petsiuk et al. (2018) | | | Metric From Walker et al. (2024) | | |
|---|---|---|---|---|---|---|
| **Test Type** | **Ins ($\uparrow$)** | **Del ($\downarrow$)** | **Ins - Del ($\uparrow$)** | **Ins ($\uparrow$)** | **Del ($\downarrow$)** | **Ins - Del ($\uparrow$)** |
| MDA $\gamma = 1$ | 0.914 | 0.214 | 0.699 | 0.834 | 0.255 | 0.578 |
| MDA $\gamma = 0.5$ | 0.916 | 0.206 | 0.709 | 0.841 | 0.243 | 0.599 |
| MDA $\gamma = 0$ (ours) | **0.916** | **0.206** | **0.709** | **0.854** | **0.243** | **0.611** |

### A.7 QUANTITATIVE $\kappa$ EVALUATION

In this section, we motivate our choice of $\kappa = 0.005 = 0.5\%$ with an ablation study. In these tests we set $\gamma = 1$ to activate the application of Eq (11). From Eq (11), $\kappa$ determines the minimum importance a patch must have to be assigned a significant attribution value. Thus, by increasing $\kappa$ an attribution's density decreases. In the paper, we choose the small value of $0.5\%$ importance such that all patches with any model importance are attributed. When $\kappa = 0\%$ every patch receives a value relative to its order. When $\kappa$ is greater than or equal to the importance of the most important patch, $A^{dense} == A^{sparse}$. In this test, we only report MAS scores (Walker et al., 2024), as changes in $\kappa$ do not affect insertion and deletion scores (Petsiuk et al., 2018).

We use $\kappa = [0, 0.1, 0.25, 0.5, 0.75, 1, 2, 3, 5, 10, 15, 20, 25]$ as percentages and we evaluate over 100 ImageNet images. Figure 9(a) plots MAS insertion - deletion as a function of $\kappa$. We see when $\kappa = 5\%$ the attribution has approached $A^{sparse}$ as the score stagnates, and when it is $0\%$, the attribution scores poorly. We choose $\kappa$ such that the attribution scores well and attributes more patches than $A^{sparse}$. In (b), we plot the model importance of the highest to lowest value patches across all 100 images. This provides insight into what portion of patches are being selected with different values of $\kappa$. Since the largest value here is 0.04 or 4%, we confirm $\kappa > 5\%$ yields $A^{sparse}$ on average over these images. Furthermore, we see $\kappa = 0.5\%$ results in Eq (11) applying to nearly $60/196$ patches. $\kappa = 0.5\%$ provides the right balance between only attributing valuable patches (the score is high) and attributing a significant number of patches (the attribution is dense). Finally, we visually confirm this behavior with Figure 10 where we see $\kappa = 0\%$ results in poor attributions with too many background attributions, increasing $\kappa$ leads to $A^{sparse}$, and $\kappa = 0.5\%$ provides the best balance for $A^{dense}$.

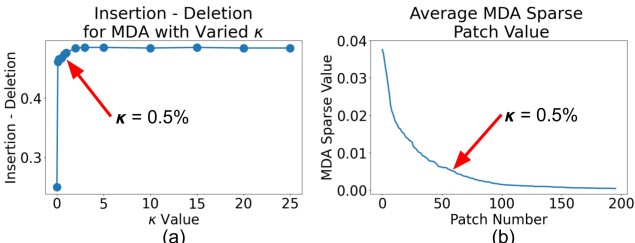

Figure 9: Quantitative comparison of MDA with varied $\kappa$. We see $\kappa = 5\%$ provides the best balance of (a) insertion - deletion score and (b) number of patches strongly attributed.

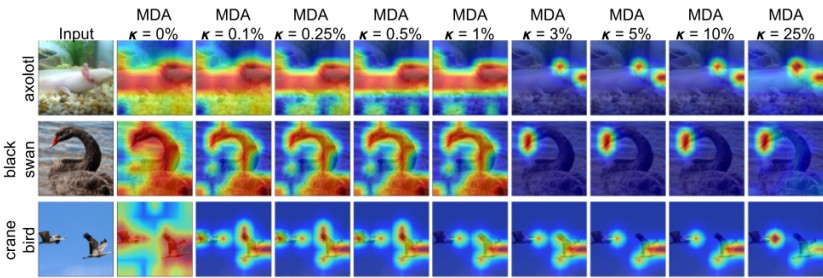

Figure 10: Qualitative comparison of MDA with varied $\kappa$. We see $\kappa = 5\%$ provides the best choice as it consistently results in attributing all important features without attributing the background.

## A.8 Evaluation of MDA vs SHAP

The Shapely value sampling attribution SHAP (Lundberg & Lee, 2017) is a popular perturbation-based method that can be considered in our patch order optimization problem. To make a comparison of our MDA method to SHAP, we would first define our approach as existing between the occlusion perturbation (Zeiler & Fergus, 2014) and SHAP (Lundberg & Lee, 2017) approaches. The occlusion method generates attributions by measuring the impact of masking only one input patch at a time. However, this results in issues where relationships between input features cannot be determined, but a $\mathcal{O}(M^2)$ runtime is achieved, where $M^2$ is the total number of patches. On the other hand, SHAP, if discussed in our context, would generate every possible ordering of the patches in the input image without regard to the metrics and then an evaluation of each of the resulting $M^2$ attributions would be made and the best scoring one would be chosen. This leads to SHAP having a runtime of $\mathcal{O}(2^{M^2})$. As a benefit of our problem definition and methodical approach, we are able to greedily select the best patch at every search step, preventing an iteration over all orderings, and yielding a $\mathcal{O}(M^2)$ runtime. SHAP has a runtime of $2.4$s compared to the $1.1$s of MDAon the ViT-base $32 \times 32$ model.

In addition to this, SHAP has significantly worse performance both quantitatively and qualitatively. When performing the segmentation test from Table 2, we see SHAP performs significantly worse than MDA in Table 14 below, losing in every test by a large margin. In addition, in Figure 11 below, we see that MDA consistently provides attributions which are both focused on the interesting features of the image subject and do not focus on the unimportant background features, while SHAP struggles with localization and has significant background attributions.

Table 14: SHAP vs MDA on segmentation test for the ViT-base $32 \times 32$ model

| Metric | MAP | IoU | F1 |
|---|---|---|---|
| SHAP (Lundberg & Lee, 2017) | 0.648 | 0.512 | 0.405 |
| MDA (ours) | **0.796** | **0.702** | **0.487** |

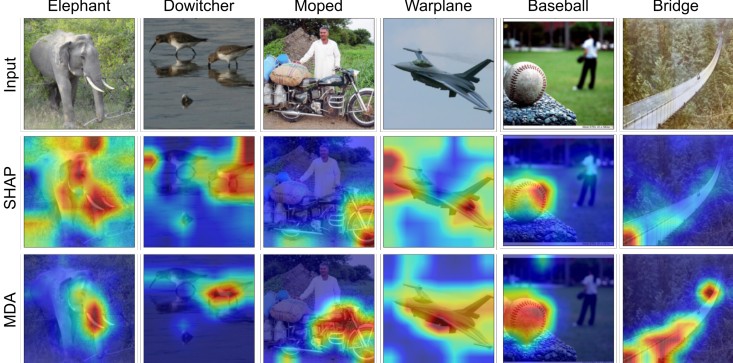

Figure 11: A visual comparison of MDA with SHAP. We see clear improvements by MDA over SHAP in all examples. MDA shows reduced background attributions and better subject localization.

## A.9 Extended Qualitative $\gamma$ Evaluation

First, we provide detailed qualitative evaluation of the visual effects as MDA attributions move from sparse to dense with varying $\gamma$ across all three models. We perform this analysis in Figure 12 for the ViT-base $16 \times 16$ model in (a), the ViT-base $32 \times 32$ model in (b), and the ViT-tiny $16 \times 16$ model in (c). We show the results for 4 images per model as $\gamma$ transitions from 0 to 1 by steps of 0.1. For each model, we present a mixture of images with small and large subjects. In Figure 12 (a), we see the two images with small subjects of class "rifle" and "flute" have consistent attributions for all $\gamma$. This behavior indicates, for these images, the model requires all of the subject's features for classification and no less. For the other three images, we see the attributions across the subjects grow in intensity and density with increasing $\gamma$, indicating only a subset of the subject's features provide a decision. In

(b) we see identical behavior to (a), however, we see a larger change between low and high $\gamma$. This could be explained by the larger patches used by the $32 \times 32$ model which contain more information. Lastly, in (c) we observe the change in attribution density with increase in $\gamma$ is much less pronounced for the ViT-tiny model. This could be attributed to the fewer layers in the model which results in learning less specific features, thus more features of any subject are needed for a decision to be made.

This study shows the effectiveness of MDA for creating both sparse and dense attributions. Additionally, we see studying MDA attributions with varying $\gamma$ can show behaviors otherwise unseen through analysis of current attributions which lack sparsity control. MDA can provide an understanding of the minimum required information a model needs to make a decision and how this varies with model parameters, which can be valuable for choosing the best model for a given application.

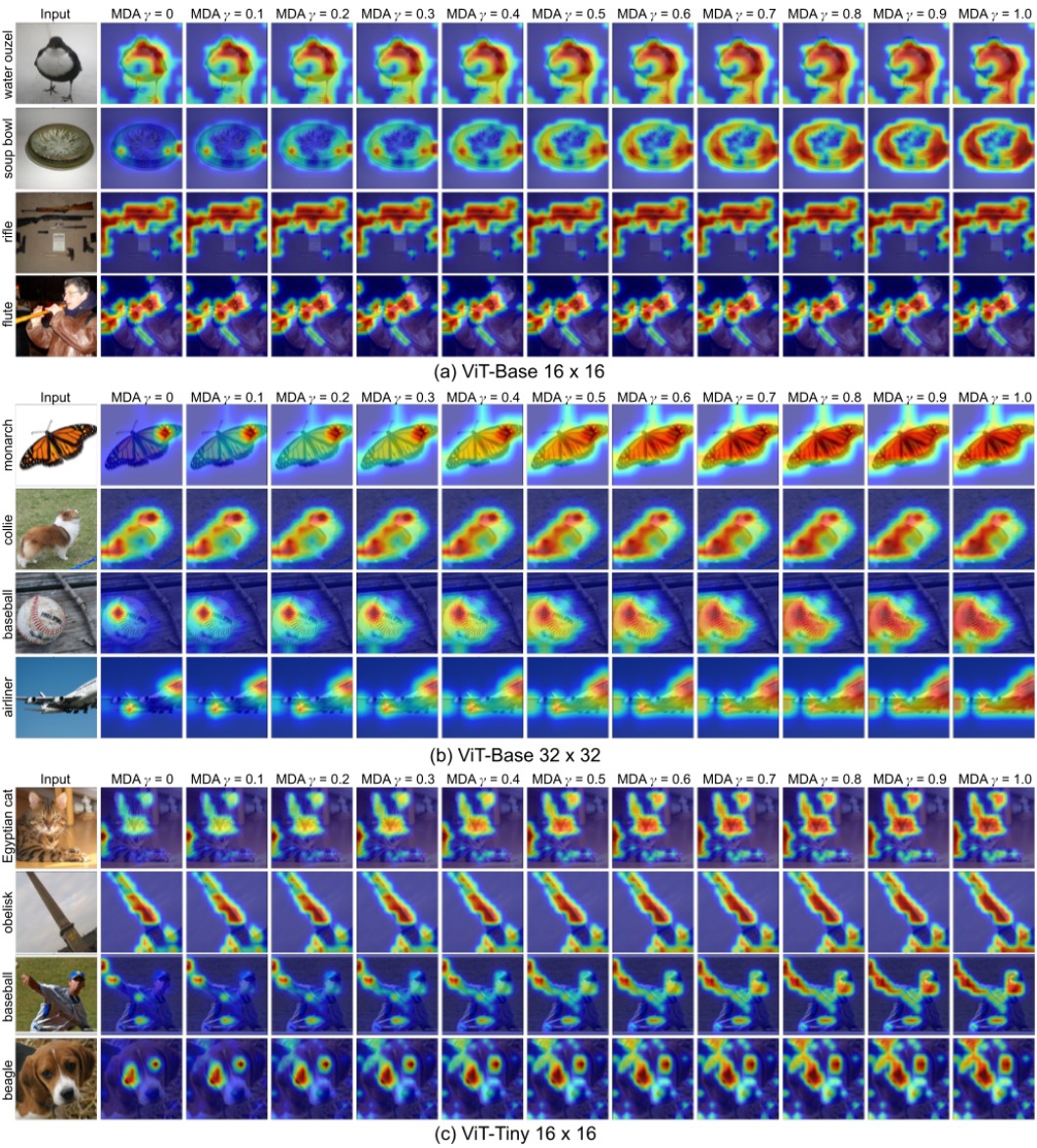

Figure 12: We analyze how changes in $\gamma$ effect the sparsity to density transition of an MDA attribution. We see for the first two ViT-base models (a) and (b) that the sparsity to density transition is smooth and significant, meaning the models find many features important, but only need a few. However, the minimal transition for ViT-tiny (c), indicates the model requires more features for a decision.

## A.10 QUALITATIVE EVALUATION OF MDA SEED ATTRIBUTION

MDA as presented and evaluated in this work uses a high-quality input attribution to reduce its search space. We present a small selection of examples to indicate the impact this "seed" attribution has on the final MDA result. In Figure 13, we present three examples which include the following: an input image, GC (Selvaraju et al., 2017), IG (Sundararajan et al., 2017), T-Attn (Yuan et al., 2021), T-Attr (Chefer et al., 2021a), ViT-CX (Xie et al., 2023), TIS (Englebert et al., 2023), and Bi-Attn (Chen et al., 2023) attributions, the output of MDA with each of these attributions as an input, labeled as "MDA seed", and MDA without a seed. All examples are for the ViT-base $32 \times 32$ model and the images are from ImageNet. In the first example (a) we see a wide variety in the input attributions. GradCAM provides no attribution on the bird subject, IG provides attributions on a large portion of the image, and the remaining methods provide attributions on the bird with varying degrees of background attribution. However, we see each MDA attribution seeded by these inputs remains both fairly consistent, and loyal to MDA without a seed. This is desirable, as the input attribution is shown to not destroy the performance of MDA . In (b) we see the same behavior as before. The input attributions vary widely in appearance and quality, yet the MDA outputs maintain a level of consistency, although reduced from the consistency of (a). We see this pattern continue in example (c). Overall, MDA with a seed provides the benefit of greatly reduced runtime without a marginal loss in quality. MDA is best used with a high-quality attribution, and future higher-quality attributions could provide a better seed for better results.

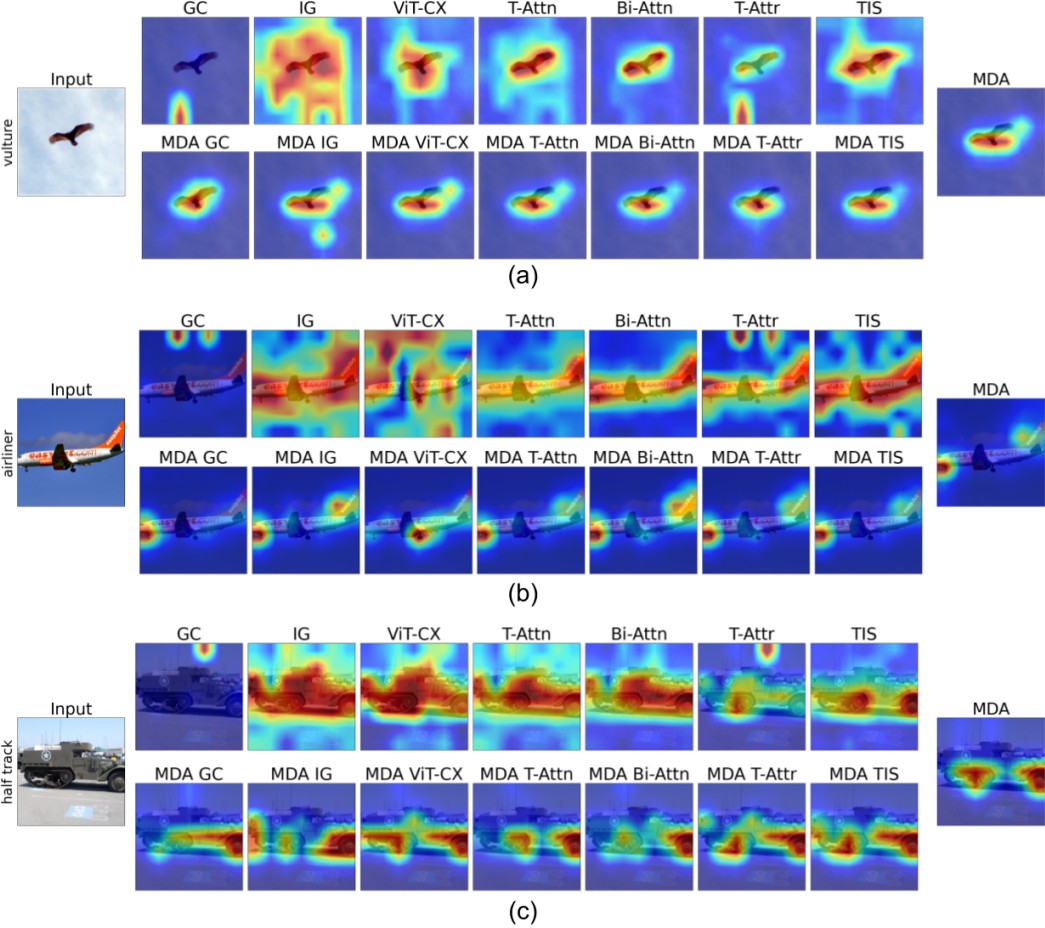

Figure 13: We qualitatively compare the performance of MDA with various "seed" attributions used as input against MDA without a seed. Regardless of seed quality, MDA provides an output consistent with the unseeded version, indicating the input attribution does not have a large impact on final result.

### A.11 EXTENDED QUALITATIVE VISUAL PERFORMANCE EVALUATION

We now present a large selection of attribution comparisons generated from the ImageNet validation dataset which extend the comparisons in Figure 6. We contrast MDA with $\gamma = 0$, $\gamma = 0.5$, and $\gamma = 1$ against IG (Sundararajan et al., 2017), GC (Selvaraju et al., 2017), T-Attn (Yuan et al., 2021), T-Attr (Chefer et al., 2021a), Bi-Attn (Chen et al., 2023), TIS (Englebert et al., 2023), and ViT-CX (Xie et al., 2023). The examples are broken in to three sections, in groups of five pages. From page 23 to 27, we display attributions for the ViT-base $16 \times 16$ model. Pages 28 - 32 contain attributions for the ViT-base $32 \times 32$ model. Lastly, pages 33 - 37 contain attributions for the ViT-tiny $16 \times 16$ model. Across 15 pages we present 255 unique images. MDA presents the most consistent, high-quality attributions across the images and models. The transition from $\gamma = 0$ to $\gamma = 1$ consistently results in a move towards a dense attribution when the model does not require all features for classification.

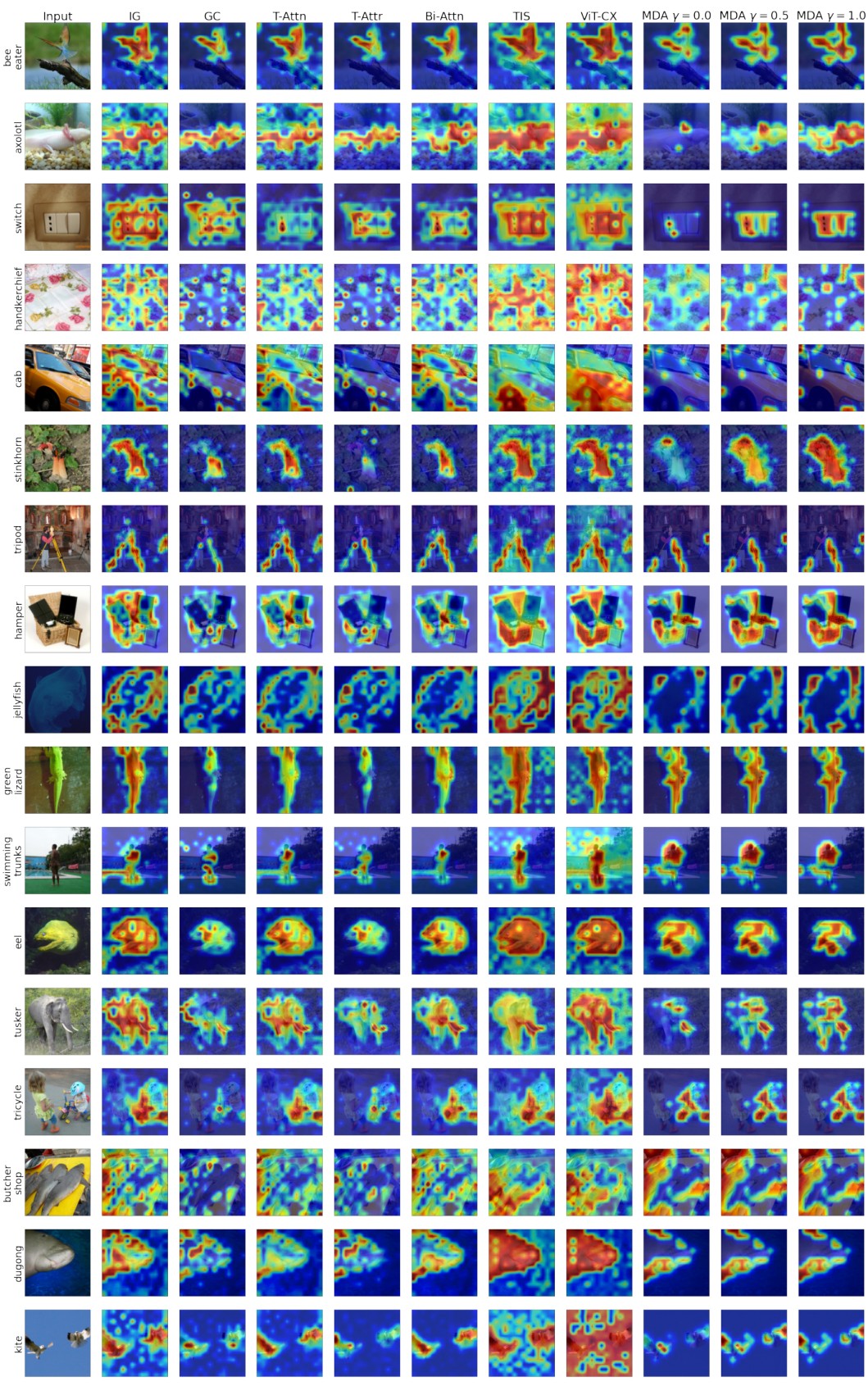

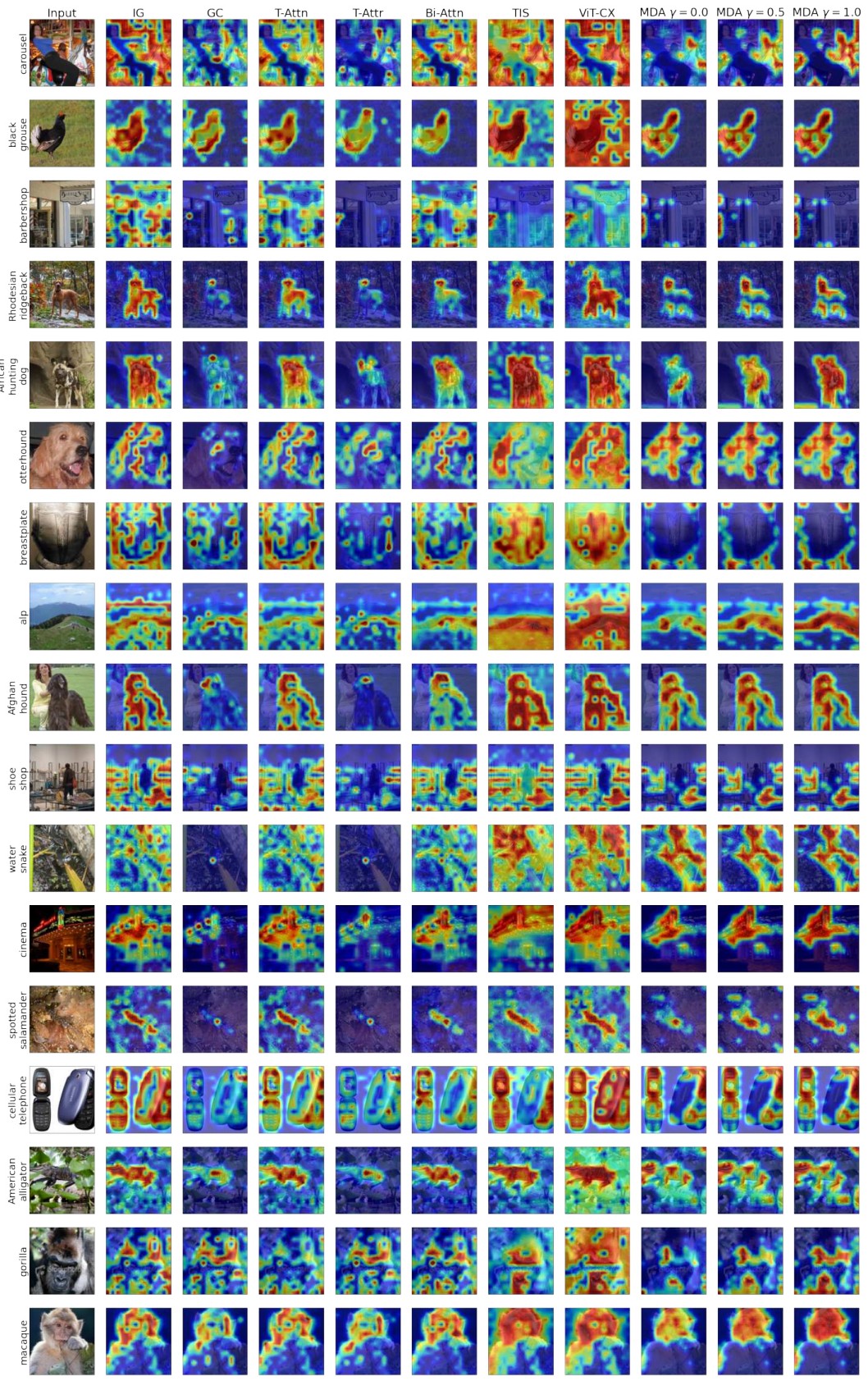

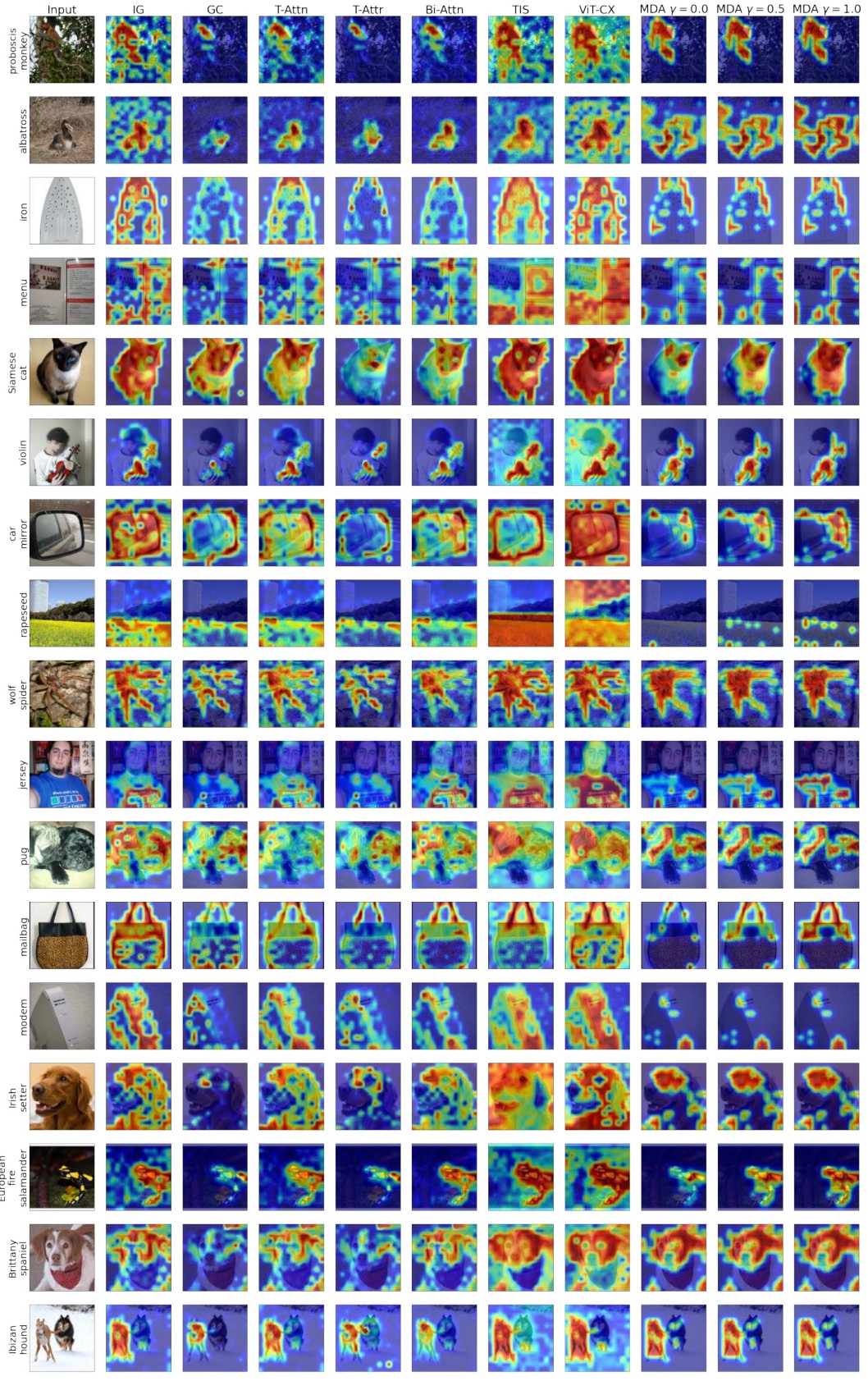

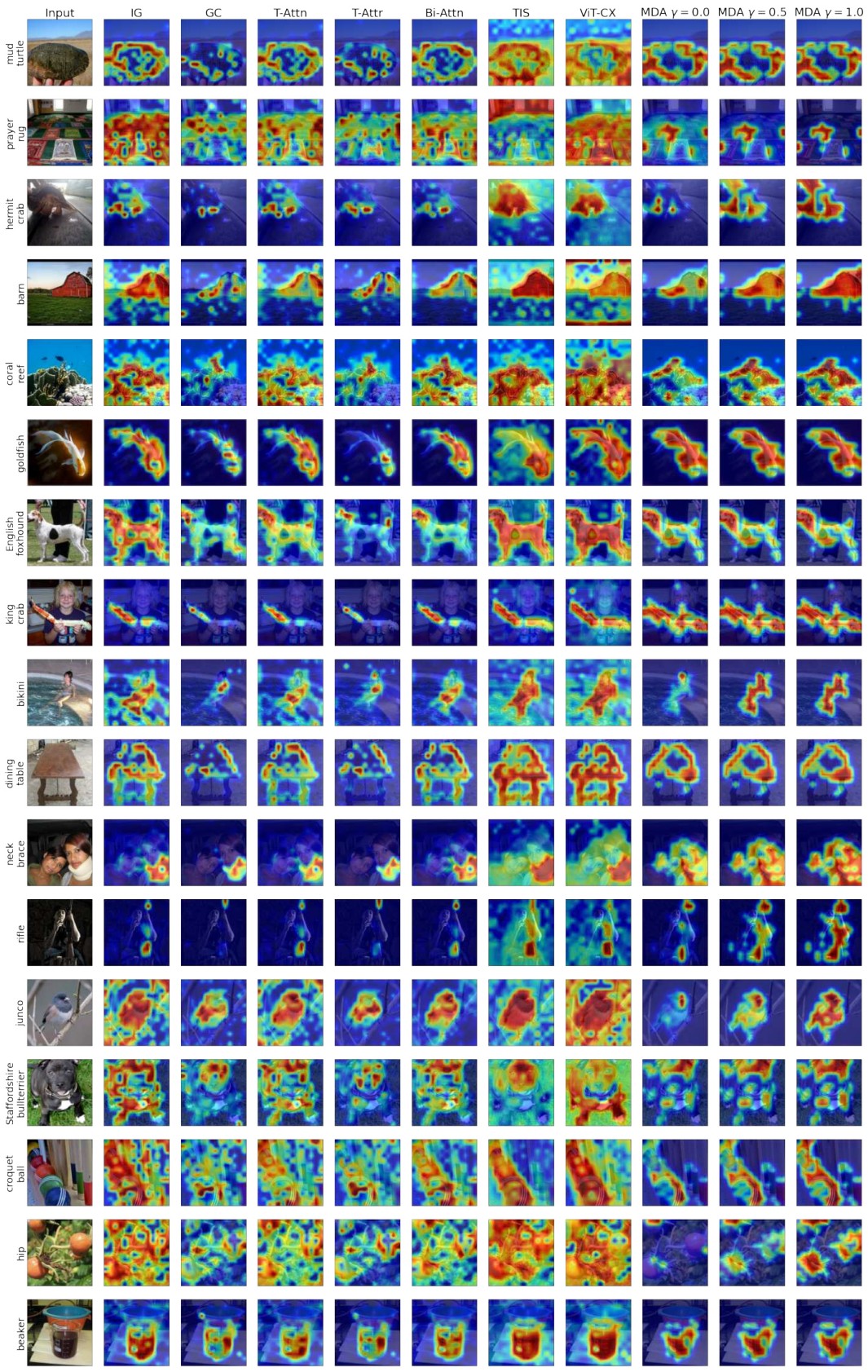

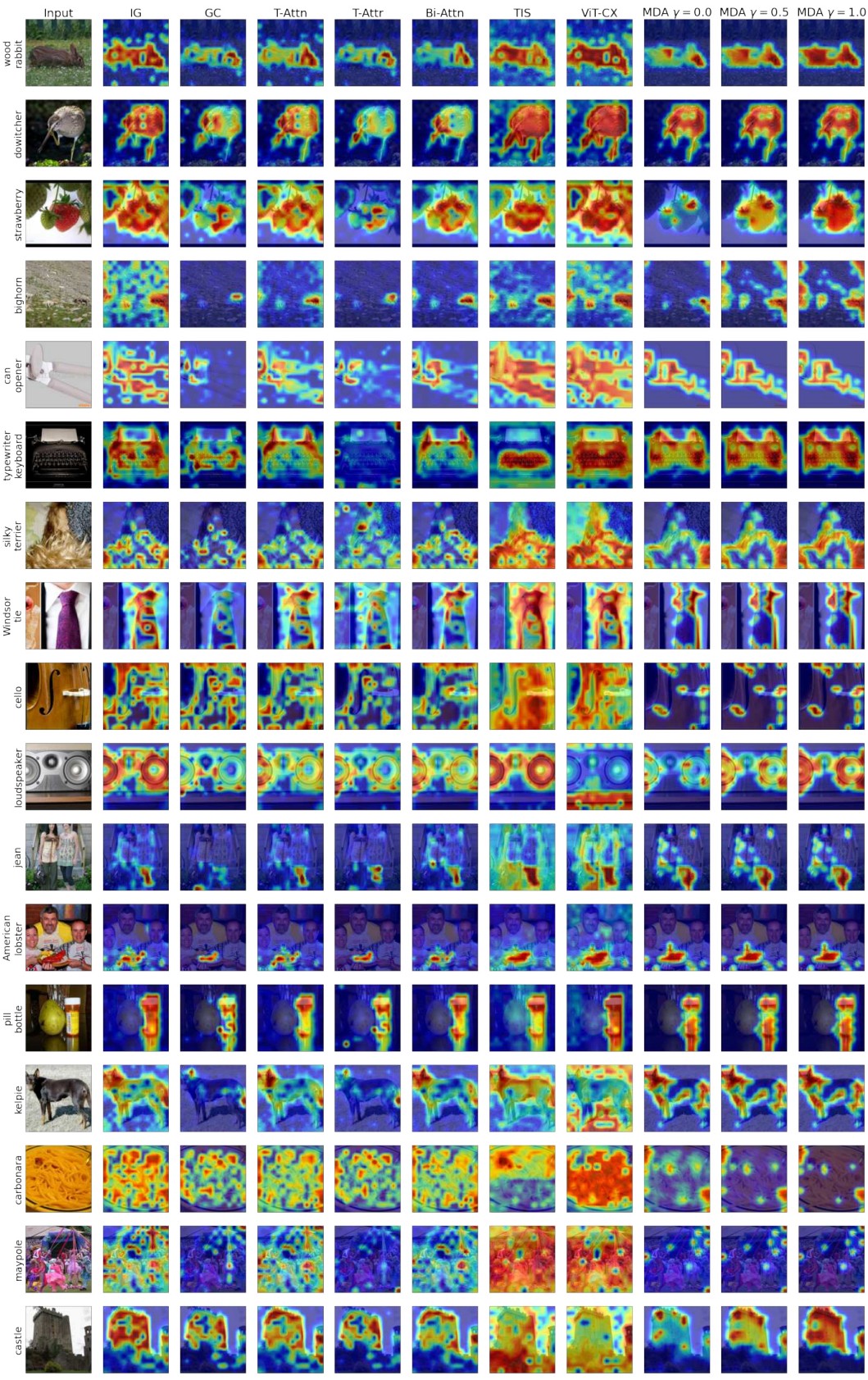

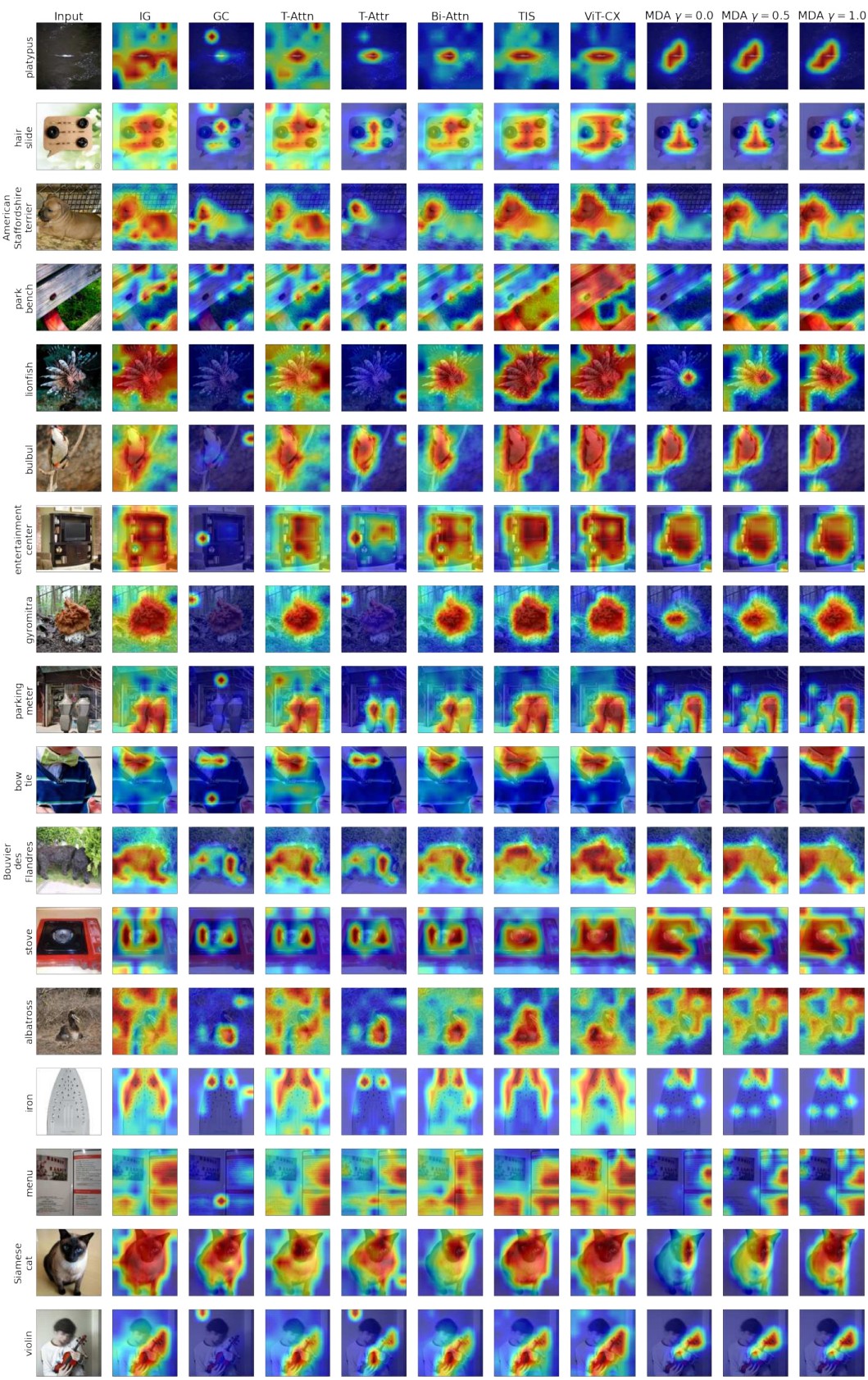

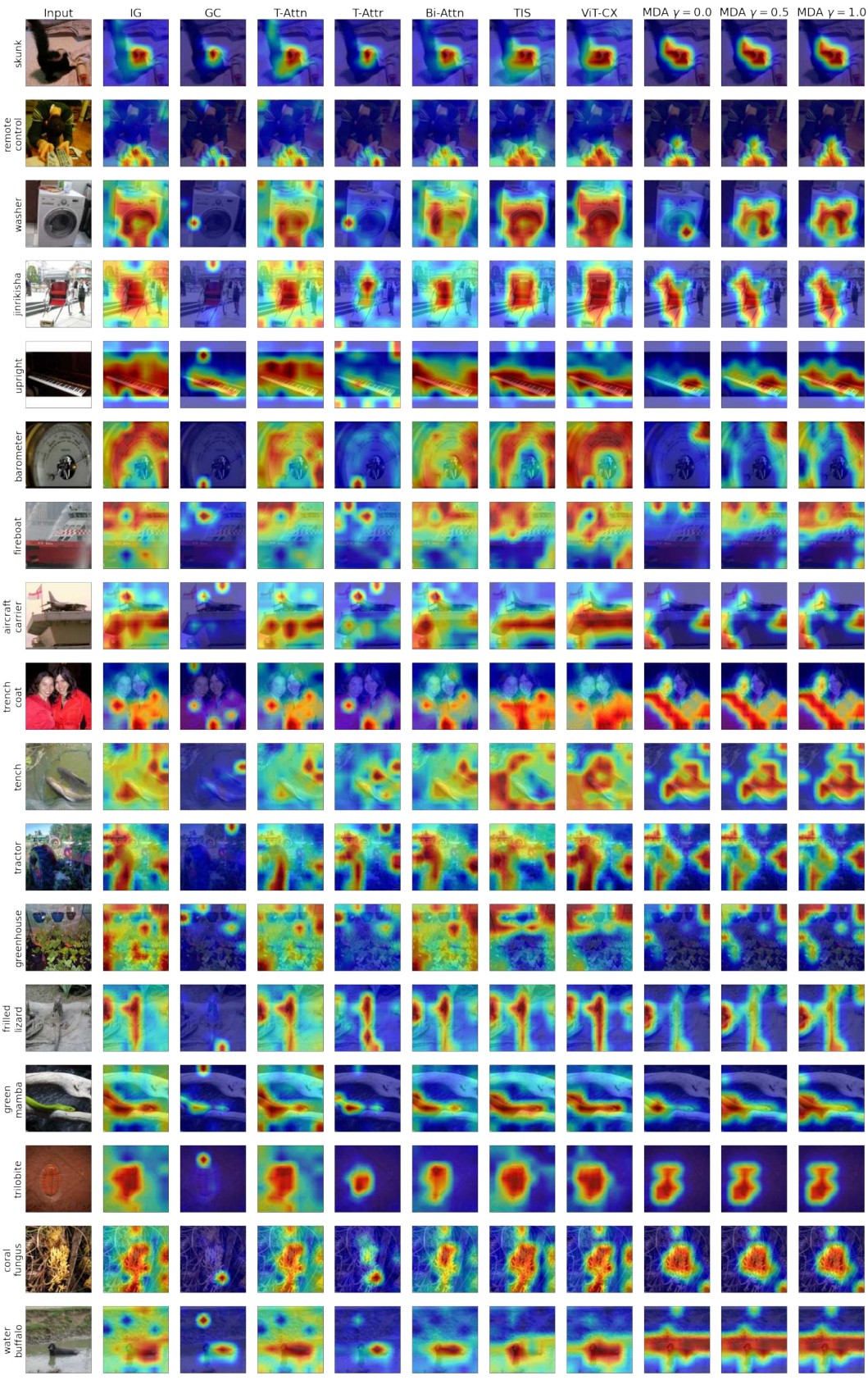

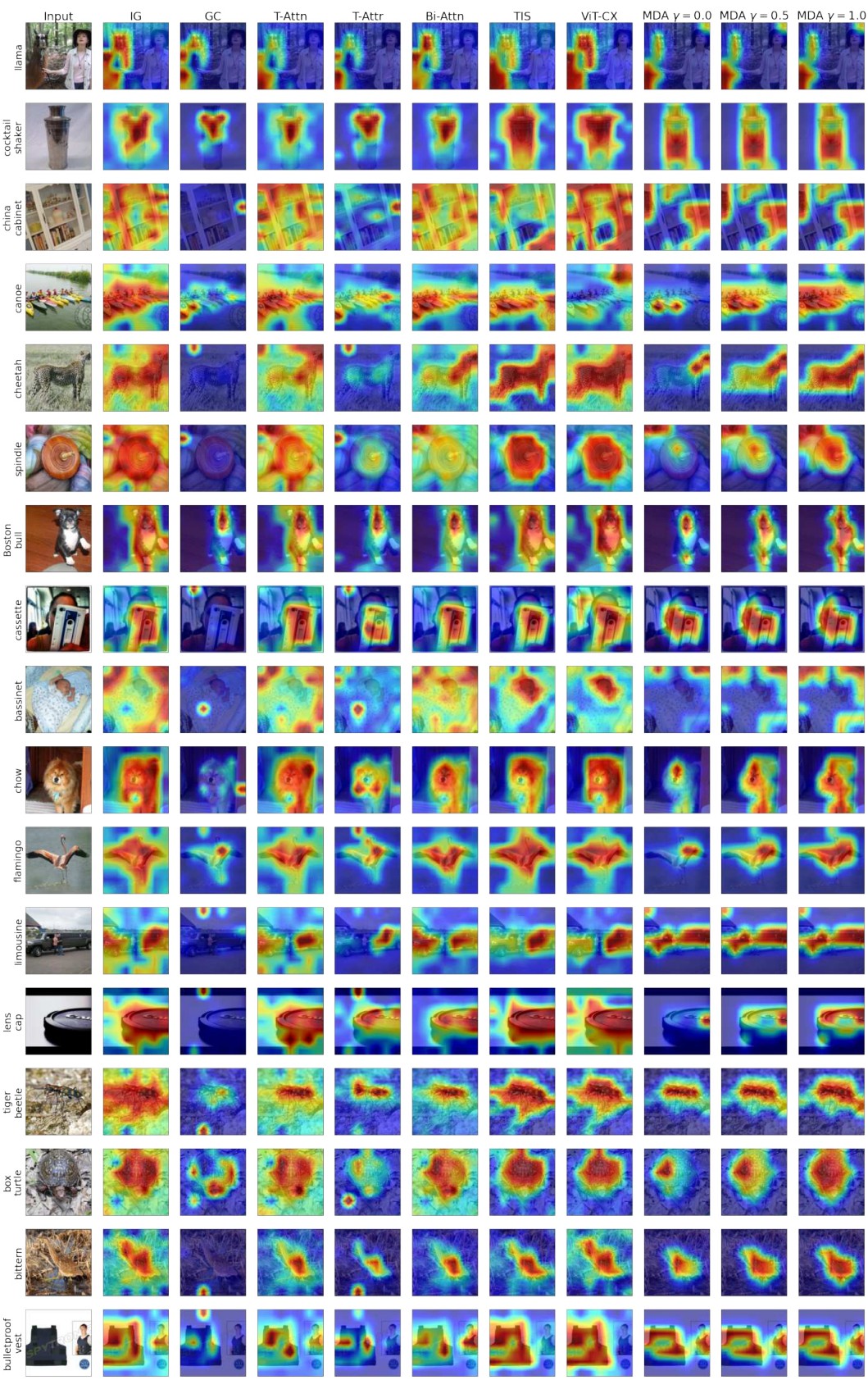

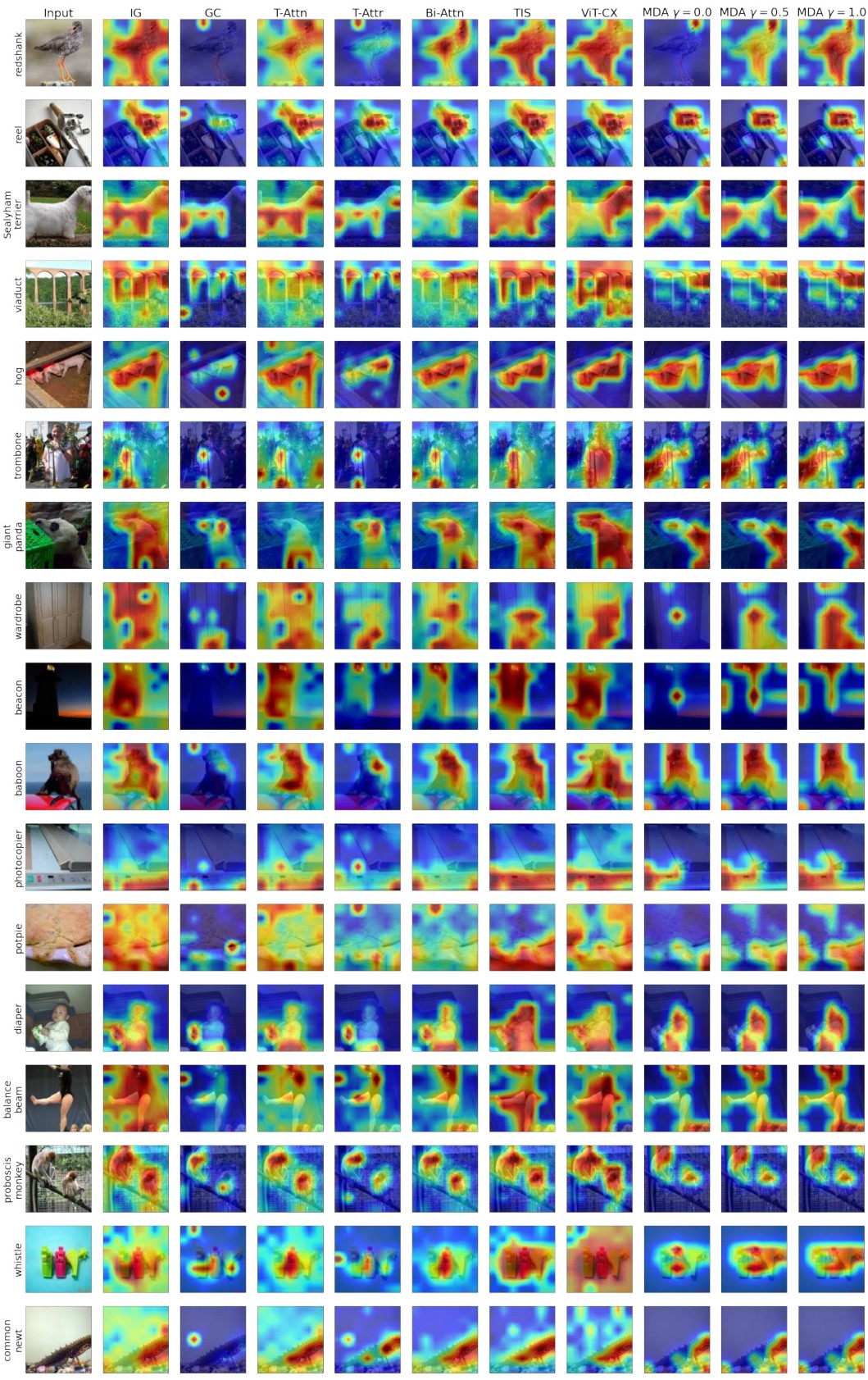

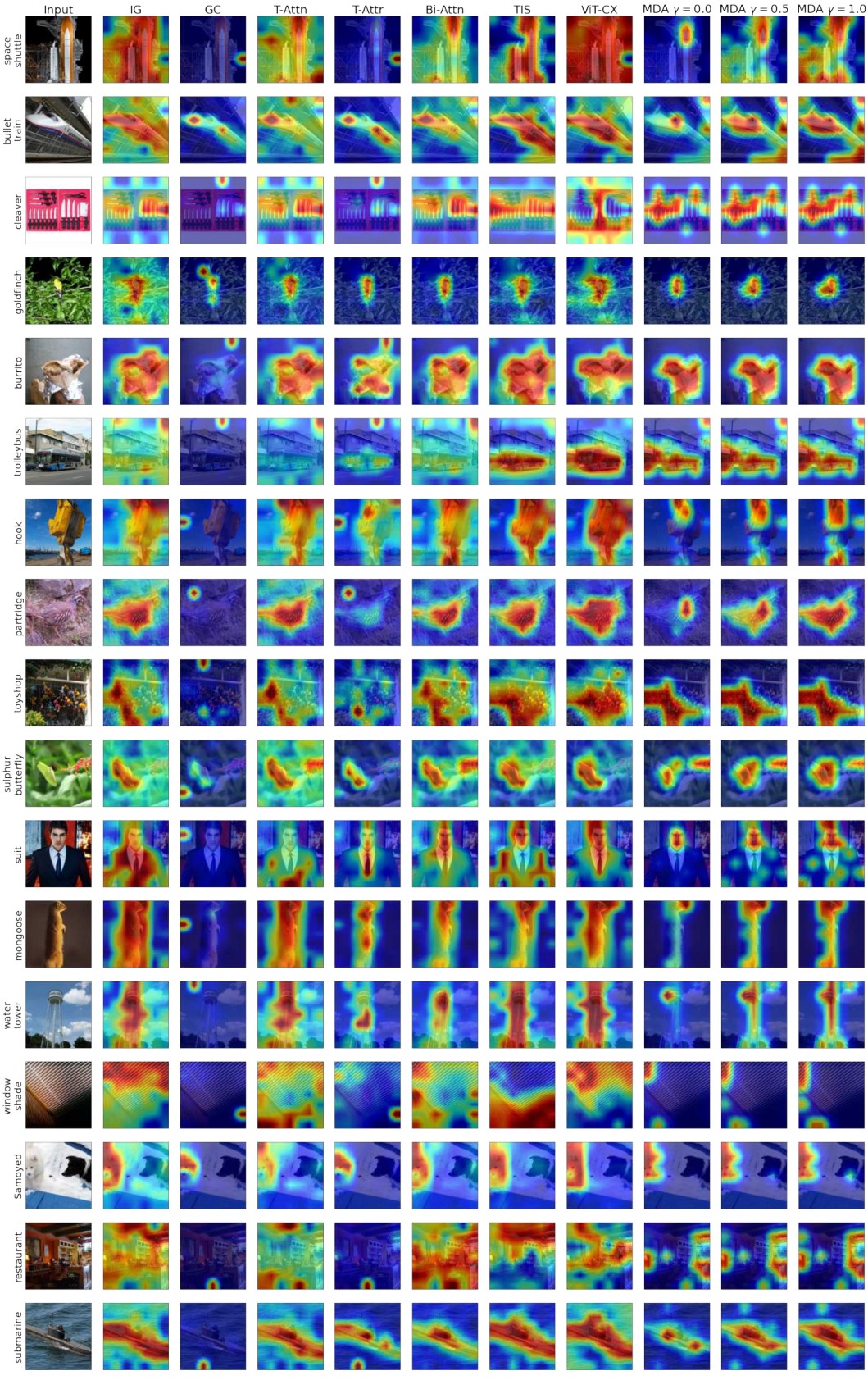

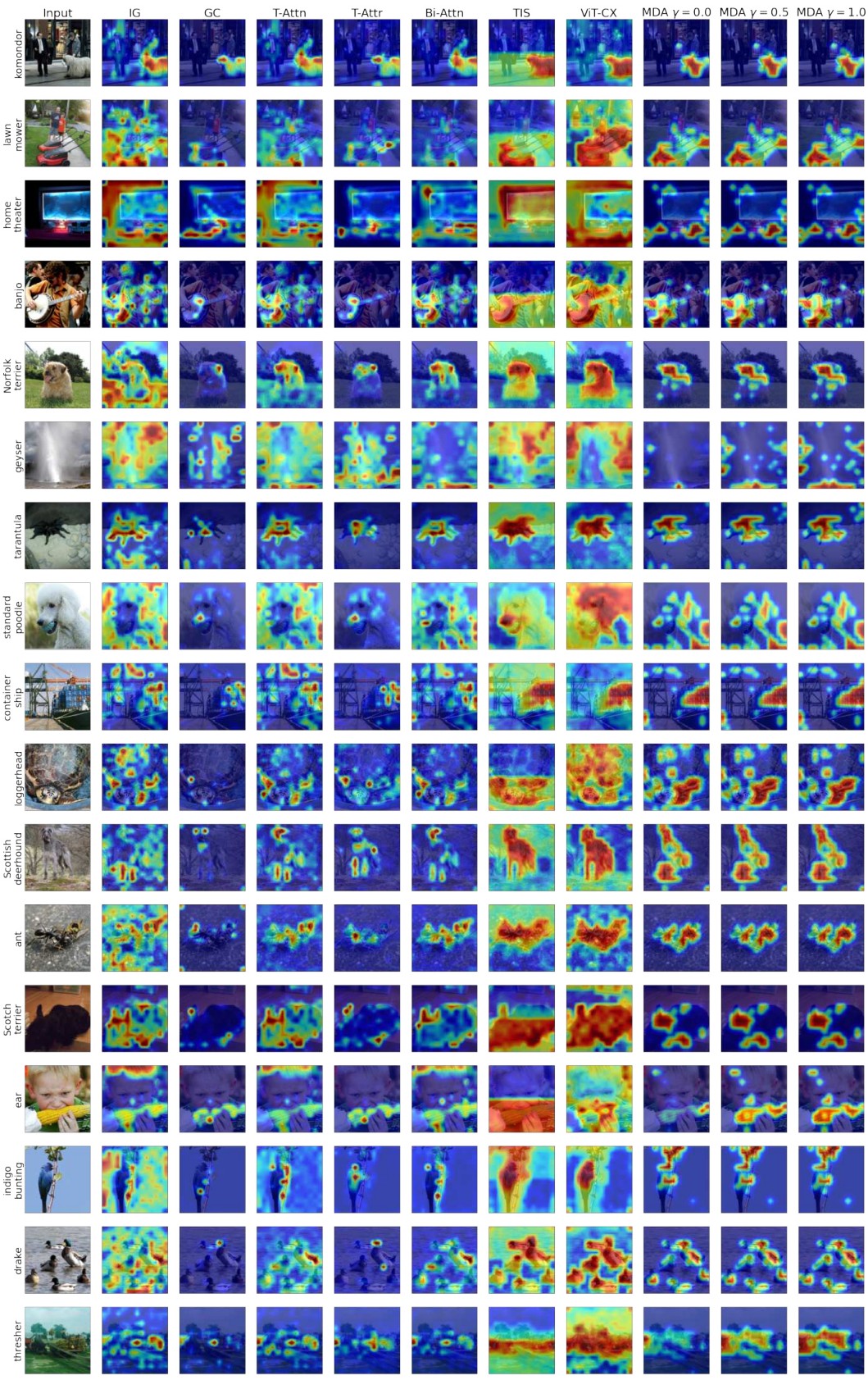

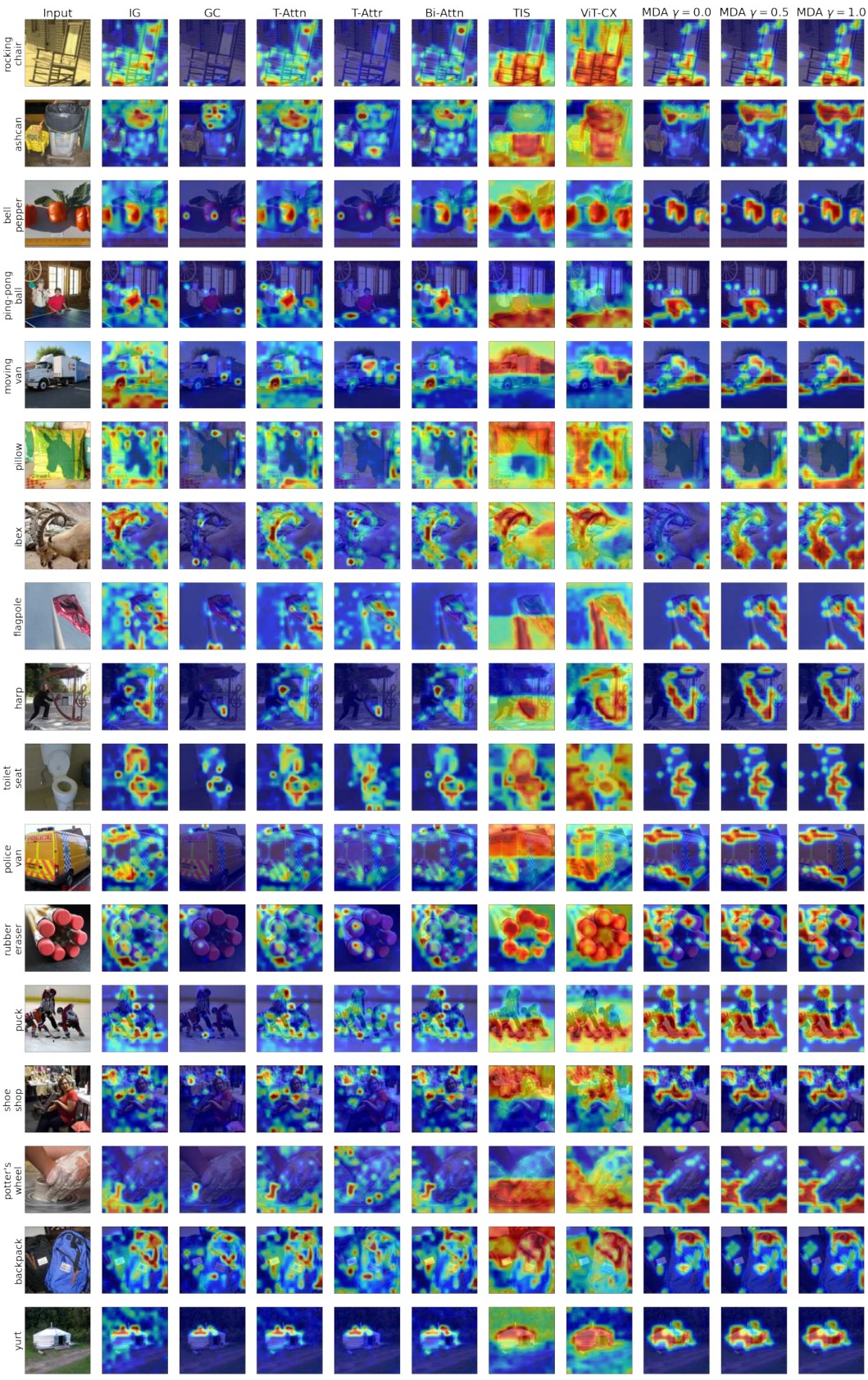

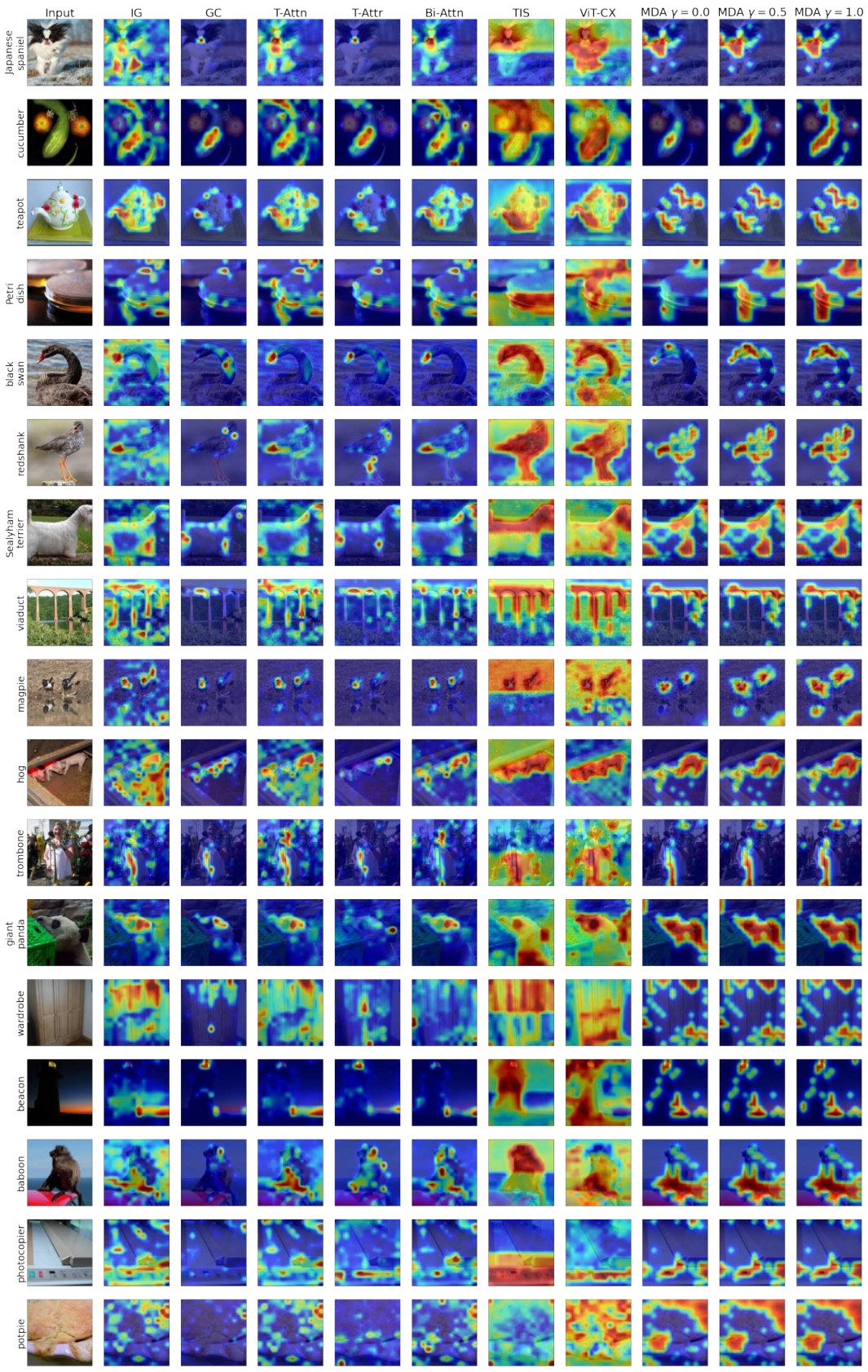

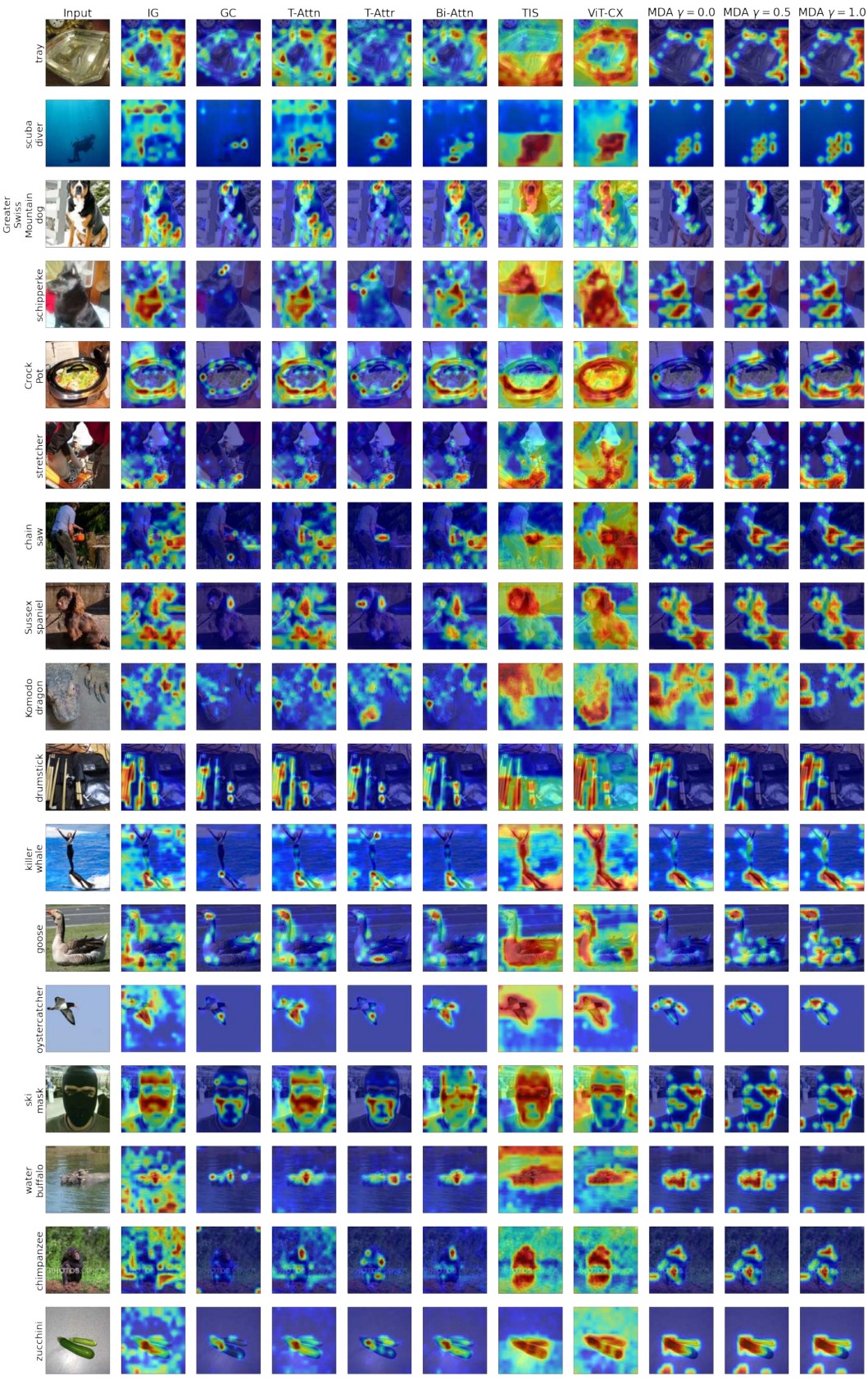

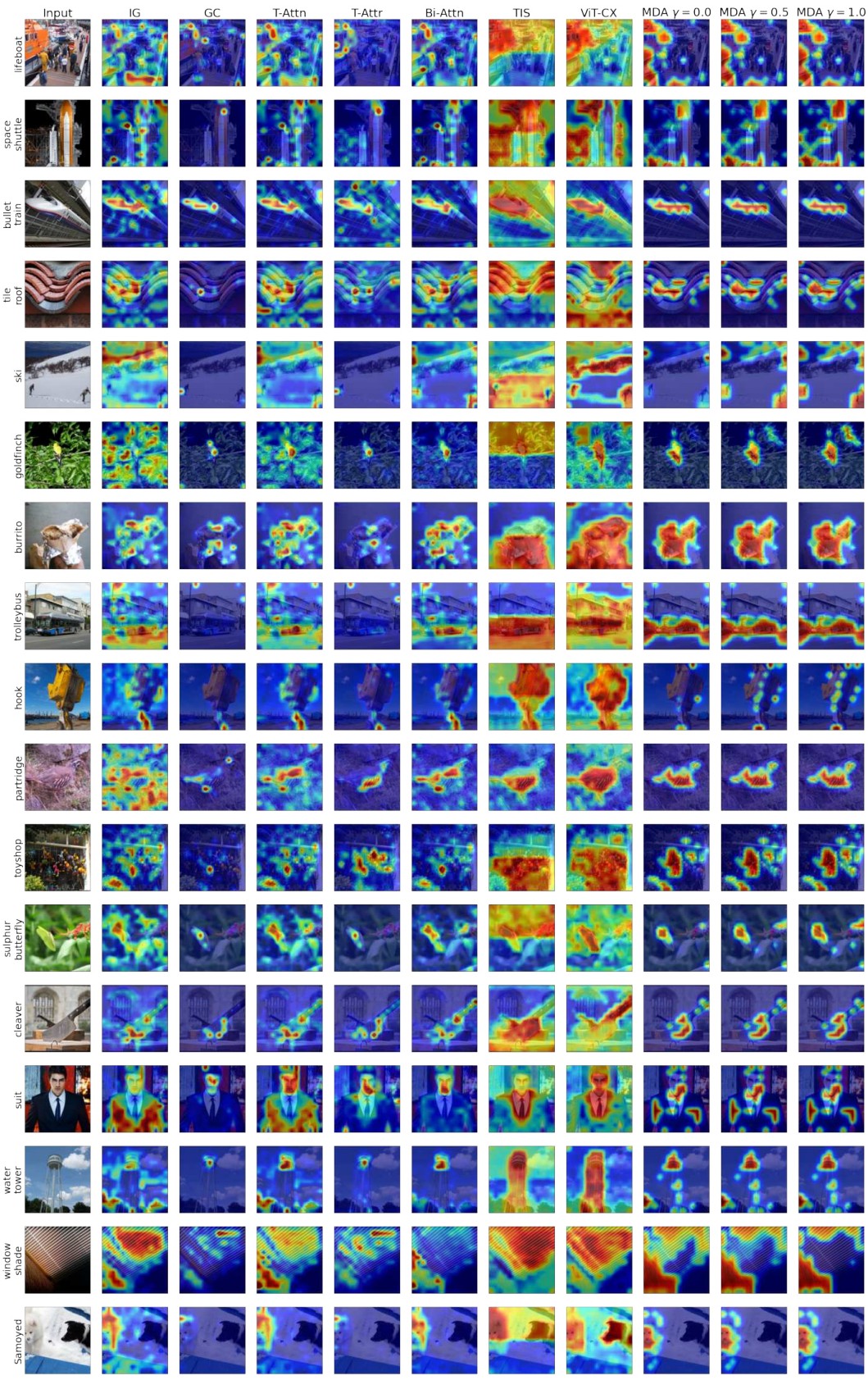

