# OpenReview forum: "Metric-Driven Attributions for Vision Transformers"
_ICLR.cc/2025/Conference — ICLR 2025 Poster_

### Official Review · Reviewer_DmfJ · 2024-10-22

**Soundness:** 4
**Presentation:** 2
**Contribution:** 3
**Rating:** 6
**Confidence:** 3

**Summary:**

The authors are dealing with Explainable AI subject dedicated to vision transformers. Their idea is to optimize the order of patches based on the insertion-deletion tests. This is done firstly by convert a blurry image into a sharp one and at each time-step to find the most prominent patch (until a given limit). Then, the order of the rest of patches is found by imitating the deletion process. Lastly, a process to set a magnitude per each is described.

**Strengths:**

1) The idea of optimizing based on the examining metric is interesting, and sounds novel to me.
2) The non-technical parts are fluent and well written.
3) considering time complexity is important when applying XAI in online real-time applications. I think it is important to pay attention to this when developing a method (even for explainability which consider mostly as offline)

**Weaknesses:**

1) I find it problematic that the approach uses the same metric for derivation and experimental evaluation. To show agnosticity for the evaluation metric, it is much needed to evaluate in a clean setting when the metric is not involved in the derivation process (or actually optimized by it). Two additional fair potential metrics for evaluation in addition to the one presented in the paper could be: segmentation test and perturbation test. (elaborated on the T-Attr paper by Chefer et al.)
2) The technical parts are hard to follow, and sometimes indigestible. 1) in line 123 the notations $MR^{ins}$, $MR^{del}$ were used before definition. I would recommend define them first before using them. Moreover, in equation (3) you defined "$\|\cdot\|$ is the sum of all selected values" (line 129) - where it appears twice in the equation and in general overlaps with the norm symbol. I would recommend not shortening it and write it explicitly: $\Sigma_{i=1}^{M}\|A_i\|$ where $M$ is total number of attributions.
3) Some evaluations are missing: a) to better understand its effect on attribution sparsity or metric scores, ablating $\kappa$ would have been complete the ablation nicely. In a reasonable scale - something like 0 to 20-25 percentage. b) time efficiency comparison is missing and might shed light on the speed-up claims in section 3.4. (did not find also in the Supp.). A complete test comparison should include the timing of several other prominent approaches, however, I am aware that it might be difficult in the rebuttal frame-time. Thus, to have a better sense of timing, it would be nice to measure the time required to obtain explainability using your method for a single image (better to report it with std of several runs).

**Questions:**

1) As I understand it, you are selecting an equal number of elements at each time-step ($\frac{D^2}{N}$). Could you provide an intuition why?

2) How would you suggest tuning the $γ$ parameter for an unknown image?

3) Explainability could be beneficial in debugging misclassified images. It would be nice if you provide examples of misclassified cases and demonstrate how MDA addresses them.

Overall, I find similarities between your approach and general spectral analysis. When moving from a blurry to a sharp image, selecting the subset of pixels that most impact classification at each time-step is somewhat corelated with examining the image’s spectrum. I believe this is good direction for further research that has potential.

I still decided to give this paper a minor negative review due to two main concerns: (1) The method should be evaluated with metrics other than ins-del to ensure a fair comparison, and (2) a comparison of time efficiency is necessary, especially given its detailed discussion in the paper.

That being said, I think the idea presented in the paper is strong, and I would be willing to upgrade my rating if my concerns will addressed.

---

> ### Author Response · Authors · 2024-11-22
>
> Reviewer DmfJ, thank you for identifying clarity issues with our definitions, and raising questions about paramater selection. Here is our response.
>
> > Q1: In line 123 the notations for $MR^{ins}, MR^{del}$ were used before definition. I would recommend defining them before using them.
>
> We have not been able to identify this issue in the text. Can you please provide more details so we may find this issue and correct it
>
> ---
>
> > Q2: In Eq (3) you defined "$|.|$ is the sum of all selected values" (line 129) - where it appears twice in the equation and in general overlaps with the norm symbol. I would recommend not shortening it and writing it explicitly: $\sum_{i=1}^M|A_i|$ where $M$ is total number of attributions.
>
> Thank you for pointing out the confusing notation of Eq (3). We have updated the equation to provide clarity and remove the ambiguity of its original definition.
>
> ---
>
> > Q3. Can an ablation study on $\kappa$ be shown?
>
> We have updated the supplementary material to include a $\kappa$ ablation evaluation as requested. Please see **Appendix A.6** for the reasoning behind our selection of the $\kappa$ parameter. Through qualitative and quantitative analysis in this section, we reinforce that $\kappa = 0.5\%$ is the best selection for generating $A^{dense}$.
>
> ---
>
> > Q4. As I understand it, you are selecting an equal number of elements at each time-step ($D^2/N$). Could you provide an intuition why?
>
> If this is in reference to the definition of the insertion and deletion tests on **line 93**, we are following the original authors in their definition of the tests [1]. They opt to perform perturbations of groups of pixels instead of single pixels, as selecting a group of pixels is more likely to select a feature than a single pixel would. If this is not what you meant, can you please provide clarity?
>
> ---
>
> > Q5. How would you suggest tuning the $\gamma$ parameter for an unknown image?
>
> The selection of $\gamma$ for unknown or unseen images will be dependent on the desired application of the attribution. *For downstream tasks such as confidence or out-of-distribution measures, we would recommend $\gamma = 0$*, as we show a lower gamma will provide only the most important features, and thus the most important model information. *For human-facing model evaluation, we would recommend $\gamma = 1$* as it may be preferable for human understanding. However, MDA with $\gamma = 0$, $\gamma = 1$, or any value in between can be calculated at the same time for nearly zero cost. Due to the design of the algorithm, the $\mathcal{O}(M^2)$ ordering process is performed only once, and the patch values are assigned for $\gamma = 0$ and $\gamma = 1$ without model calls. MDA $\gamma = (0,1)$ is then an interpolation between these sparse and dense attributions. Therefore, *MDA $\gamma = 0$ and $\gamma = 1$ can generated for any unseen image without meaningful computation increases* and thus an evaluator can have both sparse and dense interpretations of the model decision if necessary.
>
> ---
>
> > Q6. Explainability could be beneficial in debugging misclassified images. It would be nice if you provide examples of misclassified cases and demonstrate how MDA addresses them.
>
> Thank you for this suggestion. There is a great amount of existing literature which explores the use of attributions for confidence measures [2], out-of-distribution detection [3], and model debugging [4]. In this paper, we focus solely on the computation of attributions as is common in the literature, and thus we decided to focus our revisions on the key points which question our method's quality. However, we think this is an interesting future MDA application that should be explored.
>
> ---
>
> > S1. Overall, I find similarities between your approach and general spectral analysis. I believe this is good direction for further research that has potential.
>
> Thank you for this suggestion. We will take into consideration the relationship between MDA and spectral analysis when researching applications of MDA in the future.
>
> [1] RISE: Randomized Input Sampling for Explanation of Black-box Models, BMVC 2018 \
> [2] Attribution-Based Confidence Metric For Deep Neural Networks, NeurIPS 2019 \
> [3] GAIA: Delving into Gradient-based Attribution Abnormality for Out-of-distribution Detection, NeurIPS 2023\
> [4] Explanation-based human debugging of nlp models: A survey, MIT Press 2021

---

> > ### Comment · Reviewer_DmfJ · 2024-11-23
> >
> > Q1: very minor: In the revised pdf in Line 98. I think that a short clarification sentence that superscripts ins and del stands for insertion and deletion would close this nicely.
> > Q4: I see, I was curious about the selection of an equal number of pixels per each step (and of course not necessarily a single pixel). I find it interesting to split it into groups with variable sizes at each step, as at each unblurring step, it might be that selecting subset of different size will yield better results, depend on the frequency information of the input image I guess. I understand that this is borrowed from the original test.
> > Thanks for your detailed answer.

---

> > > ### Author Response · Authors · 2024-11-24
> > >
> > > Thank you for clarifying the conversation about Q1. We have made modifications in the text to improve the clarity.
> > >
> > > We are happy to have helped with the reasoning behind the step size choice of this work as discussed in Q4. It would be interesting to explore the use of different step sizes in the future design of attribution metrics.

---

### Official Review · Reviewer_7fM8 · 2024-10-29

**Soundness:** 3
**Presentation:** 2
**Contribution:** 3
**Rating:** 6
**Confidence:** 4

**Summary:**

The paper is interested in a new attribution method, specifically designed for Vision Transformers to maximize the insertion and deletion metrics used for evaluating attribution methods in the literature. The method, called Metric-Driven Attribution (MDA), is separated in two stages. The first stage finds the best order of importance among all the patches, by gradually inserting patches and evaluating each time the model response. The second stage then assigns an attribution magnitude to all patch that will preserve the order found in the first stage. An additional hyperparameter is finally introduced in the attribution magnitudes, to govern the sparsity of the magnitudes. The model is then compared against different attribution methods applied to ViT models, measured using the same insertion and deletion metrics.

**Strengths:**

- **Originality**: The underlying idea of the paper of proposing method that measures attribution score with insertion and deletion is sounded and novel.
- **Clarity**: The organization of the paper and description of the method are clear. The paper also includes multiple well-made figure that helps understand the overall method.
- **Significance**: The method provides both qualitatively and quantitatively good attribution maps. Attribution maps are usually the type of explanation methods that aligns the most with human preference in terms of explanation. The modular design of the method is also flexible to be further improved.

**Weaknesses:**

- **Significance**: The major downside of the method is the runtime, as discussed in section 3.4. The runtime depends on both the granularity of the patch decomposition and the complexity of the model to analyze, in a multiplicative way since the method requires multiple forward pass to the model, from $\mathcal{O}(M^3)$ to $\mathcal{O}(M^4)$ depending on if the search space is pruned, where $M^2$ is the total number of patch. This can make the computation of the explanation for even a single image very expensive, but it is not compared with the other methods in the experiments. Furthermore, the method is currently limited to ViT models.
- **Quality**: The metrics considered for quantitative evaluation are only the ones optimized for (insertion/deletion AUC scores). It would be also informative to include a comparison with other types of metrics based on having ground-truths, such as evaluating for the segmentation task using ImageNet-Segmentation, as done in the evaluation of other methods (in Bi-Attn or T-Attr methods for instance).
- **Clarity**: The description of the metrics in section 2.1 is not really clear, and it is an important part to understand the method and the paper. Equation 2 does not correspond to the area under the MR curve, directly summing all $MR_k$ would not give the AUC. Furthermore, line 97 it is said "MR curve is formed on the range $[0,1]$", but it is not clear what variable is ranging in $[0,1]$ here.

**Questions:**

- Could you provide a running time comparison with the other methods ? I noticed the mean running time per image was reported for MDA, but I would be interested in a comparison.
- Could this approach be applied to other types of models (CNNs for instance) by patching the inputs similarly to ViTs ? The only part specific to ViT seems to be the separation into patches of the input images.
- I'm not sure to understand why the ordering of the patches is found using two separate processes, since the "deletion-based patch ordering" is actually found by insertion of patches. From what I understood, the only differences between the two are: i) the starting image is first a blurred and then a black image, and ii) the ordering is first found from most to least important, and then from least to most important. I doubt the overall ordering found by the two processes are different between them. Could you expand more on why these two processes ? Furthermore, why start from a blurred image for insertion and from a black image for deletion ?

---

> ### Author Response · Authors · 2024-11-22
>
> Reviewer 7fM8, thank you for identifying areas in which our definitions could be clarified and for proposing interesting questions about our method. Here is our response.
>
> > Q1: Equation 2 does not correspond to the area under the MR curve, directly summing all $MR_k$ would not give the AUC.
>
> Thank you for identifying the issues with Eq (2). We have corrected this such that Eq (2) properly calculates the AUC of the model response curve.
>
> ---
>
> > Q2: Line 97 says the "MR curve is formed on the range $[0,1]$". What variable is this range  referencing?
>
> By stating the "MR curve is formed on the range $[0,1]$", *we are discussing the $MR_k$ values which form the $MR$ curve*. In this line specifically, we are referencing the $MR^{ins}$ curve. We are explaining that this curve begins at $0$ and grows to $1$ as each $MR_k$ is determined throughout the perturbation test process. In addition, for $MR^{del}$ we use the $[1, 0]$ notation to describe that this curve begins at $1$ and decreases to $0$ as the perturbation test progresses.
>
> ---
>
> > Q3: Can this approach be applied to other models?
>
> This approach can, in theory, be applied to other models as *it only requires the softmax scores of the predicted model class*. For a CNN model, the input would be separated into patches just like the ViT models, but the ideal patch size for each CNN model variation would have to be considered. This is in contrast to ViT models which have an obvious choice of patch size due to their architecture which is not present for CNN models. More realistically, *this approach would be valuable to apply to other token-based models* such as BERT where each token in the input is treated as a patch to order. We think this question is valuable to consider for future work with and exploration of the MDA method.
>
> ---
>
> > Q4: Why is the patch order found using two separate processes?  I doubt the overall ordering found by the two processes are different between them.
>
> We use two processes to find the ordering of the patches because *the orders which optimize the insertion and deletion scores ($\mathcal{I}^{ins}$ and $\mathcal{I}^{del}$) are not guaranteed to be the same, and are often not*. We provide proof of this by observing the results presented in **Figure 7 and 8 as well as Tables 7, 8, and 9 in Appendix A.3**. If insertion and deletion always yielded the same order, we would see no difference in the scores with respect to the choice of $\tau$ which changes the balance of insertion-only or deletion-only optimization. In addition, please see the newly added **Figure 8** and its associated text in **A.3** which provides visual proof and discussion of how the order varies between the two optimizations.
>
> ---
>
> > Q5: From what I understood, the only differences between the two are: i) the starting image is first a blurred and then a black image, and ii) the ordering is first found from most to least important, and then from least to most important. Is finding the "deletion-based patch ordering" therefore the same process as insertion?
>
> We understand and recognize that the insertion and deletion processes both resemble an insertion process, but this for good reason, as we have shown a solo insertion optimization is not ideal. Beginning with insertion optimization, we follow the oringinal design of this test to find an order which maximizes the model response at each step. Because insertion measures the minimum features needed for classification, this is easy to solve. On the other hand, deletion, by definition, will ideally remove patches (replace them with black) from the original image which influence the *largest* change in model response first. For the reasons stated in **Section 3.1**, `"Deletion-Based Patch Ordering", it is challenging to find this order, so *we perform an inversion of the deletion test*. By inserting patches to a black image and selecting the patches which influence the *lowest* change in model response, *we are still computing deletion, but in a reverse order*. Once we invert this resulting ordering, we have done the equivalent of a deletion test.
>
> ---
>
> > Q6: Why start from a blurred image for insertion and from a black image for deletion?
>
> For the implementation of these processes, we chose the blurred image and black image baselines for insertion and deletion as *this are how the tests are defined by their original authors* [1] as explained in **Section 2.1**. Since we aim to optimize for these tests, *using the same baselines allows us to have proper evaluation of the model response*.
>
> [1] RISE: Randomized Input Sampling for Explanation of Black-box Models, BMVC 2018

---

> > ### Comment · Reviewer_7fM8 · 2024-11-25
> >
> > Thank you for the detailed answer.
> > Q4-Q6: I understand the idea of using the same baselines than [1] to start from a blurred image for insertion and a black one for deletion. However, this doesn't help to understand if the two processes would lead to the same ordering. In the experiments behind Table 7-9, and Figure 8, are you also using these two different starting setups ? I.e. are you starting from a blurred image when finding the ordering for insertion and a black one for deletion ?
> > You could consider starting from a black image, and finding patches that influence the largest change in model response when inserting them. This would be the same process as the one to find "insertion-based" ordering, but starting with a black image instead of a blurred one.
> > What would be the order in this case compared to the "deletion-based" ordering ?

---

> > > ### Author Response · Authors · 2024-11-26
> > >
> > > > Q1: I understand the idea of using the same baselines than [1] to start from a blurred image for insertion and a black one for deletion. However, this doesn't help to understand if the two processes would lead to the same ordering. In the experiments behind Table 7-9, and Figure 8, are you also using these two different starting setups ? I.e. are you starting from a blurred image when finding the ordering for insertion and a black one for deletion ?
> > >
> > > Yes, all of the experiments in Tables 7-8 and Figures 7 and 8 have been performed using the same setup as described in the definitions of the optimizations. Insertion has been performed with respect to a blurred baseline and deletion has been performed with respect to a black baseline. As mentioned by the reviewer, we made this selection to follow the evaluation metrics in [1]. We have updated the text in paragraph 1 of A.3 to make this clarification.
> > >
> > > > Q2: You could consider starting from a black image, and finding patches that influence the largest change in model response when inserting them. This would be the same process as the one to find "insertion-based" ordering, but starting with a black image instead of a blurred one. What would be the order in this case compared to the "deletion-based" ordering ?
> > >
> > > We now understand your question more clearly. You are asking for us to contrast (a) insertion ordering and (b) deletion ordering while not varying the background. We have updated Appendix A.3 paragraph 3 and Figure 8 to include a case study on this topic. We illustrate the patch orderings which result from the three ordering processes (insertion, deletion, and joint insertion-deletion) with both black baseline and blurred baseline selections. Additionally, we show the result of the joint optimization performed with insertion using the blurred baseline and deletion using the black baseline, as proposed. It can be observed that every combination of the ordering process and baseline yields a \textit{unique} patch ordering even when the same baseline is used between insertion and deletion. We additionally see that choosing a baseline which matches the evaluation metric (blurred for insertion and black for deletion) produces the best scores.

---

### Official Review · Reviewer_Q9o9 · 2024-11-03

**Soundness:** 3
**Presentation:** 4
**Contribution:** 2
**Rating:** 6
**Confidence:** 3

**Summary:**

This paper argues that attribution quality metrics should be used to also generate the attributions and proposes Metric-Driven Attribution (MDA) for explaining Vision Transformers. Based on the metric, it creates attribution maps by performing patch order-magnitude optimization across the tokens. Patches are ordered according to their importance and then assigned magnitudes. MDA is claimed to provide a smooth trade-off between sparse and dense attributions by modifying the optimization objective.


Post discussion final comments:
The authors have addressed my concern regarding optimizing and evaluating the same metric. The additional experiments are convincing. While I still believe that some applications can benefit from more reliable attributions even if they are slow, I also support arguments of reviewer DmfJ and reviewer 7fM8 that this method will not easily scale to very large architectures.

In my opinion, this paper would be of interest to the research community but the rating still stands at 6. Hence, I am maintaing my score.

**Strengths:**

The paper gives a sound motivation for its contribution and the proposed approach makes intuitive sense.

The paper is well written and easy to follow.

The proposed method can control the density of attributions.

Better quantitative results are achieved and the qualitative results also look impressive.

**Weaknesses:**

I think not optimizing the attribution maps w.r.t. the metric (that measures the attribution quality) is intentional rather than a neglect in the current literature because there is an inherent risk of biasing the results if we use the metric itself for computing the attribution maps. Provided the metric is perfect, this is a reasonable idea. However, metrics for evaluating attribution methods are still an active research topic. I would have to wait and hear the opinion of the other reviewers on this matter.

The proposed method is computationally very expensive as the insertion and deletion processes are both iterative leading to O(M^4) complexity. The proposed method reduces the search space from O(M^4) to O(M^2) but that is still significant given 14x14 patches = 196. Moreover, the search space reduction relies on using another existing attribution method, making the solution less than elegant.

Real time performance is an important criterion for attribution methods as one may require this information for every decision made by the model. Using insertion-deletion game for evaluating attribution methods, on the other hand, is an offline process so it makes more sense to use it for that purpose.

During insertion (page 4), due you use the strongest response with respect to the ground truth? Maybe give a precise definition of model response to avoid confusion.

Typos :
“may violated” -> may be violated
“memoization”  -> memorization

**Questions:**

Please see the Weaknesses section and write your response to the queries I have raised.

---

> ### Comment · Reviewer_Q9o9 · 2024-11-22
> **Metric for optimization**
>
> I see that the other reviewers also share my concern regarding the same metric being used for optimization and evaluation. This is a major limitation. Besides this, the computational complexity is also a significant limitation.

---

> ### Author Response · Authors · 2024-11-22
>
> Reviewer Q9o9, thank you for identifying areas for clarity improvement. Here is our response.
>
> > Q1: During insertion (page 4), due you use the strongest response with respect to the ground truth? Maybe give a precise definition of model response to avoid confusion.
>
> In our method, there is no ground truth. The patch order is found strictly from the measured model response. Our patch ordering processes are performing the insertion or deletion tests as defined in the related work (Section 2.1), and thus at each step we calculate the model response via Eq (1) to indicate the value of the potential patch choices. Let us clarify this process for insertion. At each step $k$, we are starting from the image $P_k$ with $k$ patches inserted. From this image, we create $N - k$ unique images where each has $1$ of the currently unused $N - k$ patches inserted. We measure the $MR$ of each of these images via Eq (1), and the image with the highest $MR$ indicates that it had the best patch inserted. *There is no ground truth available or used for this process.*

---

> > ### Comment · Reviewer_Q9o9 · 2024-11-24
> > **Patch order**
> >
> > Thank you for clarifying this. Perhaps an explicit statement in the paper would be good that the patch order is strictly from the measured model response.

---

> > > ### Author Response · Authors · 2024-11-24
> > >
> > > Thank you. We have added this clarification on line 208.

---

### Official Review · Reviewer_zur2 · 2024-11-03

**Soundness:** 2
**Presentation:** 2
**Contribution:** 2
**Rating:** 3
**Confidence:** 5

**Summary:**

This paper studies the attribution-based ViT explanation methods. The authors propose using attribution
quality metrics to generate the attributions as explanation results. The designed method, Metric-Driven
Attribution (MDA), consists of 2 steps: The first step orders the patches in terms of importance and the
second step assigns the magnitude to each patch while preserving the patch order, which can generate
smooth results. Experiments on ImageNet indicate the superiority of MDA.

**Strengths:**

1. The research question "why are the metrics not used to generate the attributions?" is interesting.
2. The two-step method can reduce the runtime from O(M^4) to O(M^3).

**Weaknesses:**

1. Motivation.

$\Delta$-DiT: Accelerating Diffusion Transformers without training via Denoising Property Alignment | OpenReview
The reviewer's main concern is the motivation of this work. The proposed method MDA is exclusive to
perturbation-based evaluation metrics, such as Ins, Del, Ins-Del, etc. However, perturbation-based
evaluation only represents a specific aspect of desirable properties of explanation results, especially
considering that there are no ground truths for explanation (otherwise, we can adopt ground truths as a
method to generate explanation results). Therefore, the contribution of this work seems insignificant and
even questionable, as the proposed MDA ignores other evaluation metrics, such as localization ability [1],
faithfulness [2, 3], visual quality [1], sanity check [4], etc.

[1] transformer interpretability beyond attention visualization. CVPR 2021.

[2] Rethinking Attention-Model Explainability through Faithfulness Violation Test. ICML 2022.

[3] on the faithfulness of vision transformer explanations. CVPR 2024.

[4] Sanity checks for saliency maps. NIPS 2018.

2. Computational complexity.

Although MDA reduces the search space from O(M^4) to O(M^3), it may still be slower than other
explanation methods, especially those not based on attributions. A detailed discussion about the
computational/time complexity compared to more methods would strengthen the paper.

3. Comparison to methods based on Shapley Value.

The idea is similar to Shapley Value [5, 6]. Could the author discuss the advantages and unique
contributions of MDA in terms of eﬀectiveness and complexity?

[5] Shap-CAM: Visual Explanations for Convolutional Neural Networks based on Shapley Value. ECCV 2022.

[6] A Unified Approach to Interpreting Model Predictions. NIPS 2017.

**Questions:**

As shown in Tables 1 and 2, the proposed MDA method exhibits performance degradation on Del ("Metric
From Petsiuk et al. (2018)"). Is there any analysis of this phenomenon? Is this related to how the patch
order is determined in step 1?

---

> ### Author Response · Authors · 2024-11-22
>
> Reviewer zur2, we thank you for your valuable questions and commentary. Here is our response.
>
> > Q1: Why did you choose to use perturbation-based metrics to optimize? Why are metrics such as localization ability, faithfulness, visual quality, sanity check, etc. ignored?
>
> Thank you for this excellent question. There are indeed many different explainability metrics that could have been selected for optimization. The MDA framework could easily be adapted to optimize many of these alternative metrics. However, we selected to optimize the insertion and deletion metrics [1] and their MAS extensions because *they are the most popular metrics in the literature*. Moreover, unlike many faithfulness metrics [2, 3, 4, 5] the score of the *MAS metric is not only dependent on the order of the attributions*, but also the magnitude, which disambiguates many different attribution maps that would otherwise score similarly but visually look very different. Furthermore, we have revised the paper to show that *optimizing for the MAS metrics gives very good results for additional explainablity metrics* such as the ImageNet Segmentation pointing game [6] (**Table 2**), negative and positive perturbation [4, 6] (**Table 3**), and monotonicity [3] (**Table 6**), where it outperforms the SOTA methods on all tests. The inclusion of these tests illustrates *MDA's ability to succeed in localization ability, faithfulness, and visual quality.*
>
> Some of the other metrics are not very practical for assessing the quality of an attribution map. For example, the Sanity Checks [7] or ROAR [8] methods require model editing or retraining and are therefore only used in a limited number of studies (while they might be cited substantially). In addition, *we aim to provide single-image explanations*, which would be not be as feasible with these methods since they often operate over an entire dataset. Due to the slow nature of performing these sanity checks, we could not include evaluation results in this response time frame.
>
> [1] RISE: Randomized Input Sampling for Explanation of Black-box Models, BMVC 2018 \
> [2] Is Attention Interpretable?, ACL 2019 \
> [3] One explanation does not fit all: A toolkit and taxonomy of AI explainability techniques, ACL 2020 \
> [4] ERASER: A Benchmark to Evaluate Rationalized NLP Models, ACL 2019 \
> [5] Rethinking attention-model explainability through faithfulness violation test, ICML 2022 \
> [6] Transformer interpretability beyond attention visualization, CVPR 2021 \
> [7] Sanity checks for saliency maps, NeurIPS 2018 \
> [8] A benchmark for interpretability methods in deep neural networks, NeurIPS 2019
>
> ---
>
> > Q2: The idea is similar to Shapley Value. Could the author discuss the advantages and unique contributions of MDA in terms of effectiveness and complexity?
>
> We appreciate you raising the discussion of the similarities between our MDA method and the popular perturbation-based SHAP method [9]. We have included a direct comparison of these two methods in the new **Appendix A.9**. From those quantitative and qualitative results, we see that *MDA is not only clearly outperforming SHAP in attribution quality, but it also has a significantly faster runtime.*
>
> [9] A unified approach to interpreting model predictions, NeurIPS 2017
>
> ---
>
> > Q3: As shown in Tables 1 and 2, the proposed MDA method exhibits performance degradation on Deletion. Is there any analysis of this phenomenon? Is this related to how the patch order is determined in step 1? Why does MDA perform worse in deletion?
>
>
> The MDA framework does exhibit weaker performance in the deletion test as shown in the tables. However, this behavior is not surprising and it is indeed due to the patch order determined by step 1. We originally discussed this behavior with our $\tau$ ablation study. We will explain this behavior with reference to **Figure 7 and Tables 7 - 9 in Appendix A.3** Since MDA optimizes for both insertion and deletion, the insertion - deletion score is maximized, but the two individual scores may be slightly worse as a result. This is illustrated in these figures and tables where we see that *optimizing only for deletion will maximize the deletion score*. Figure 7 shows that we can always select a value of $\tau$ (which balances the insertion and deletion joint optimization) that beats all tests in *deletion or insertion* if that is a desirable behavior. However, it is best to have an attribution which both tests prefer equally, rather than one which only performs well in one test. This leads to the choice $\tau = 0.9$ to balance insertion and deletion as seen in Figure 7.

---

### Author Response · Authors · 2024-11-22

Thank you to all the reviewers for their valuable feedback and great questions. We are providing an overall response to the reviewers' main concerns as well as individual responses to each of the reviewers' questions. We have also correspondingly revised the paper to include new results and descriptions to resolve these concerns. We will indicate the locations of these changes with our answers to the reviewers' questions and concerns. The common questions among reviewers were concerns for 1) the need for evaluation metrics which differ from those we optimize for (reviewers: zur2, Q9o9, 7fM8, and DmfJ) and 2) full analysis of the runtime of MDA compared against SOTA methods (reviewers: zur2, Q9o9, 7fM8, and DmfJ).

To address this first concern, we followed the suggestions of the reviewers and implemented three additional evaluation metrics: an ImageNet Segmentation pointing game, positive and negative perturbation tests, and a monotonicity test. These tests cover faithfulness, localization ability, and visual quality. We have put the first two tests (segmentation and positive/negative perturbation) in **Tables 2 and 3 of Section 4.1** of the main paper and have updated the associated text in that section. The monotonicity test is in **Table 6 in the new Appendix A.2**. *It is observed that MDA improves the performance on these metrics as well.* We believe the inclusion of these additional results should show that the MDA method's performance extends past the metric it optimizes for.

For the second concern, we have included a full runtime analysis in **Table 10 of the new Appendix A.4**. We compare MDA's execution time against all other attribution methods across all three models. While the runtime of the MDA framework is slower than for other attribution methods, it is in the range from $1.1$ to $13.3$ seconds. *This is still sufficiently fast for most applications of attribution methods* as attribution maps are often used to understand AI model behaviors after system failures. We believe that many users would value attributions of  superior quality at the expense of longer runtime. At the very least, it would be a valuable tool for many users to have in their explainablity toolbox.

---

> ### Comment · Reviewer_DmfJ · 2024-11-23
>
> First of all, thanks for the impressive additional comprehensive experiments, I think it is absolutely shed more light on the performance of your proposed approach.
> 1) Additional tests with different metrics - I think that the superior results based on common metrics are giving more credibility to the transferability of your approach. I tried to compare results with other papers (i.e. [1]) and found it difficult since the experiments are based on different architectures (you are reporting on ViT-B\32 whereas Chefer et al. report based on ViT-B\16). In general I think that compare against more architectures would be better, however, since also [1] did the same (report only on ViT-B\16) I think it is fair enough.
> 2) Timing evaluation - as you mentioned in the comment, I agree that better explanation can come in the expense of slower runtime. Nevertheless, following your additional timing experiment, MDA is order of magnitude slower than all other approaches, and I believe that this would exacerbate for larger architectures. If we take CLIP [5] as an example for a prominent architecture that ViT embedded in, then it trained on ViT-B\32, ViT-B\16 and ViT-L\14. Namely, that small and tiny are less in use in practice I believe.
>
> Minor:
> Regarding the segmentation and perturbation tests - I see that older papers are reporting the results ranging from [0,100] ([1],[2]) and newer papers are already report ranging [0,1] ([3],[4]) - I can guess that it is due to the code shared by Chefer et al. (not sure what is better, but in general common conventions are always preferable).
>
> [1] Hila Chefer et al. Transformer interpretability beyond attention visualization. CVPR 21
> [2] Tingyi Yuan et al. Explaining information flow inside vision transformers using markov chain. eXplainable  21
>
> [3] Alexandre Englebert et ak. Explaining through transformer input sampling. ICCV 23
> [4] Weiyan Xie et al. Vit-cx: Causal explanation of vision transformers. IJCAI 23
>
> [5] Radford et al. Learning transferable visual models from natural language supervision. PMLR 21.‏

---

> > ### Author Response · Authors · 2024-11-24
> >
> > Thank you reviewer DMfj for taking the time to read our revised paper and response in detail. With this work, we have proposed novel ideas for computing ViT attributions based on patch order and magnitude optimization. Based on our experiments, we have found this method yields superior results over SOTA methods - with improvements spanning a varied selection of common metrics - at the expense of longer runtime. We believe this work advances the body of science and brings to light new possibilities for neural model explanation, while providing a meaningful reference point to compare with SOTA methods. Moreover, we believe consistently improving attribution quality over numerous metrics is more challenging than achieving a fast runtime. Overall, we are glad to have resolved all of the reviewer's initial concerns. Given that the only remaining weakness of the paper is the longer run time (still in seconds for moderate size ViTs), we hope that the reviewer would consider increasing the score to a weak accept due to the quality improvements from the performed revision. However, we understand that this is ultimately a judgment call for the reviewer.

---

> ### Comment · Reviewer_Q9o9 · 2024-11-24
> **Additional results**
>
> Thank you for providing a detailed response along with extensive additional results and uploading your revised paper with the additional results. These additional results are indeed impressive and address my major concern related to optimizing and evaluating the same metric.
>
> Regarding the run time, I agree that there are some applications that do not require realtime performance and can benefit from slower and more reliable attribution methods e.g. medical applications and model diagnosis.
>
> With these changes, I think this would be a valuable paper for the research community.
>
> Minor:
> I noticed some typos here and there is the added text. Please proof read and correct.

---

> > ### Author Response · Authors · 2024-11-24
> >
> > Thank you reviewer Q9o9 for providing valuable feedback and thoroughly evaluating our responses. We are glad to see we have resolved your concerns regarding MDA's performance on additional metrics and concerns regarding the runtime performance of our MDA method. We appreciate this discussion improving the impact of our novel method which explains neural models through patch order and magnitude optimization. We additionally thank you for your affirmation of the impact and quality of this work.

---

> ### Comment · Reviewer_7fM8 · 2024-11-25
>
> Thank you for taking the time to do more evaluation, and the running time comparisons.
>
> - These additional evaluations are very valuable and better support the method.
>
> - Although I agree with the authors and reviewer Q9o9, that some applications do not require real-time processing, I also agree with reviewer DmfJ that we are talking here about order of magnitude slower. This is a major limitations that can impact the possible applications of the method. For instance, this means that such method cannot be used in conjunction of any kind of method using attribution maps at training time.
>
> That being said, I also think that the paper is overall valuable for the community.

---

> > ### Author Response · Authors · 2024-11-26
> >
> > Thank you reviewer 7fM8 for your constructive feedback and full consideration of our response. We are grateful that you appreciate the value that this paper would bring to the community. Our approach delivers attribution maps of high quality at the expense of longer runtime. We concur with your assessment that MDA should not be used within training loops but is suitable for generating post-hoc explanations.

---

### Author Response · Authors · 2024-12-03
**Summary of Discussion**

We thank the reviewers for their efforts in providing valuable feedback and participating in this discussion which has improved the value, impact, and clarity of this work. Out of the four reviewers, three (Q9o9, 7fM8, and DmfJ) participated in the discussion, ***all of which have agreed that our work is above the acceptance threshold and a useful contribution to the body of science***. We did not receive any response from reviewer (zur2) after we addressed their concerns in the revision. Their main concern appears to be that, in this work, we focus solely on perturbation-based metrics. However, we believe it is important that all areas of AI research receive their due focus, and we believe the field of perturbation-based metrics can be valuable to ViT explanations. To this end, we have shown that our **MDA method**, which creates ViT attributions optimized for perturbation-based metrics, **leads to superior visual quality and quantitative performance over a wide range of metrics when compared to the SOTA**.

---

### Meta-Review · Area_Chair_TaE2 · 2024-12-18

**Metareview:**

The paper initially received mixed reviews. The major concerns were:

1. motivation - why not optimize other metrics [zur2]
2. computationally expensive, how does it compare to other methods? [zur2, Q9o9, 7fM8, DmfJ]
3. how does it compare to SHAP? [zur2]
4. why does MDA exhibit performance degradation for deletion? [zur2]
5. there is a risk of biasing the results towards the metric, which itself is not perfect. [Q9o9]
6. search space reduction uses existing attribution methods [Q9o9]
7. need to evaluate with other metrics (besides the one optimized for) [7fM8, DmfJ]
8. can it be applied to CNNs? [7fM8]
9. presentation problems [DmfJ]

The authors wrote a response and provided additional experiments on Points 7 and 2. Three reviewers were satisfied with the response (one negative reviewer flipped to positive), and appreciated that the paper would be valuable to the community. The fourth (negative) reviewer did not participate in the discussion period, but the AC thinks that this reviewer's points were addressed well enough by the authors. The AC notes that although the runtime is long, the proposed method could still be useful for offline tasks, such as model debugging. It may also serve as an "upper-bound" for more-efficient XAI methods.

In addition, the AC would like to point out another limitation/caveat that should be mentioned in the paper: the evaluation against other XAI methods is not entirely fair since MDA has the benefit of optimizing the evaluation metric whereas other methods do not. For other XAI methods, it is possible to "optimize" the evaluation metric via selection of hyperparameters, e.g., which feature layer to use, how to convert the attribution map at that feature layer to the same size as the image, how to combine attribution maps from different layers (and how to weight them), as well as other specific hyperparameters in the XAI method.  The caveat of not entirely fair experiments should be mentioned in the paper, and the authors are encouraged to consider further fairer evaluation as future work.

All things considered, the AC agrees that this paper is valuable to the community, and thus recommends accept. The paper should be revised accordingly.

As a side note, there is a related work that selects locations to optimize a faithfulness-like metric has a similar spirit to the submission (although not applied to XAI):
[BBAM: Bounding Box Attribution Map for Weakly Supervised Semantic and Instance Segmentation, CVPR 2021].

**Additional Comments On Reviewer Discussion:**

see above

---

### Decision · Program_Chairs · 2025-01-22

Accept (Poster)